# Synapsin condensation controls synaptic vesicle sequestering and dynamics

Christian Hoffmann [1,7], Jakob Rentsch [2,7], Taka A. Tsunoyama [3,7], Akshita Chhabra[1], Gerard Aguilar Perez[1], Rajdeep Chowdhury[4], Franziska Trnka[1], Aleksandr A. Korobeinikov [1], Ali H. Shaib[4], Marcelo Ganzella[5], Gregory Giannone [6], Silvio O. Rizzoli [4], Akihiro Kusumi[3], Helge Ewers [2] & Dragomir Milovanovic [1] ✉

Neuronal transmission relies on the regulated secretion of neurotransmitters, which are packed in synaptic vesicles (SVs). Hundreds of SVs accumulate at synaptic boutons. Despite being held together, SVs are highly mobile, so that they can be recruited to the plasma membrane for their rapid release during neuronal activity. However, how such confinement of SVs corroborates with their motility remains unclear. To bridge this gap, we employ ultrafast single-molecule tracking (SMT) in the reconstituted system of native SVs and in living neurons. SVs and synapsin 1, the most highly abundant synaptic protein, form condensates with liquid-like properties. In these condensates, synapsin 1 movement is slowed in both at short (i.e., 60-nm) and long (i.e., several hundred-nm) ranges, suggesting that the SV-synapsin 1 interaction raises the overall packing of the condensate. Furthermore, two-color SMT and super-resolution imaging in living axons demonstrate that synapsin 1 drives the accumulation of SVs in boutons. Even the short intrinsically-disordered fragment of synapsin 1 was sufficient to restore the native SV motility pattern in synapsin triple knock-out animals. Thus, synapsin 1 condensation is sufficient to guarantee reliable confinement and motility of SVs, allowing for the formation of mesoscale domains of SVs at synapses in vivo.

Neurons are highly polarized cells with the axonal length being orders of magnitude larger than the diameter of a cell body. SVs are often clustered in specific three-dimensional regions within synaptic boutons along the axons[1-3]. However, SV diffusion goes against their localization[4,5]: The volume of an axon is much larger than the volume of a bouton, implying that there needs to be a mechanism allowing for SVs to accumulate and remain at the synapses.

Decades of research have established that SVs are clustered by synapsin 1, an SV-associated phosphoprotein abundant at synapses[6]. The genetic data strongly support a synapsin-dependent mechanism of SV clustering. Analyses of synapses in situ[7] and in culture[8] indicate that both the total number and the packing of SVs drop significantly in synapsin knock-out animals. Furthermore, an acute antibody-mediated perturbation of synapsin abolishes SV

[1]Laboratory of Molecular Neuroscience, German Center for Neurodegenerative Diseases (DZNE), 10117 Berlin, Germany. [2]Institute of Chemistry and Biochemistry, Freie Universität Berlin, 14195 Berlin, Germany. [3]Membrane Cooperativity Unit, Okinawa Institute of Science and Technology Graduate University (OIST); Onna-son, Okinawa 904-0495, Japan. [4]University Medical Center Göttingen, Institute for Neuro- and Sensory Physiology, Germany; Biostructural Imaging of Neurodegeneration (BIN) Center, Göttingen, Germany; Excellence Cluster Multiscale Bioimaging, 37073 Göttingen, Germany. [5]Department of Neurobiology, Max Planck Institute for Multidisciplinary Sciences, 37077 Göttingen, Germany. [6]Interdisciplinary Institute for Neuroscience, University of Bordeaux, UMR 5297, F-33000 Bordeaux, France. [7]These authors contributed equally: Christian Hoffmann, Jakob Rentsch, and Taka A. Tsunoyama. ✉e-mail: dragomir.milovanovic@dzne.de

clusters in the lamprey giant axons both at rest[9] and upon stimulation[10].

The classical view postulates that synapsins crosslink SVs together in a "scaffold" of protein-protein interactions between synapsin and its binding partners[11,12]. Yet, the exceptionally high concentrations of synapsins in boutons (i.e., 120 μM)[13] suggests them being more than just a modifiable scaffold for SVs[14]. Recent studies showed that synapsin 1 alone can cluster lipid vesicles via liquid-liquid phase separation (LLPS)[7,15]. While LLPS is an intrinsic property of synapsins, the synapsin condensates recruit SV integral proteins (e.g., synaptophysin[16], VGLUT1[17]) and soluble synaptic proteins (intersectin[7], alpha-synuclein[18]) through multivalent interactions. However, key questions remain unanswered, including what is the nature of accumulation and motility of SVs in synaptic boutons, and—just as important—does the region of synapsin 1 responsible for LLPS suffice to give rise to the specific diffusion properties and confinement of SVs in synaptic boutons? To address these questions, we scrutinize the dynamics of synapsin 1 and SVs both in a minimal reconstitution system and in living neurons capitalizing on the (ultrafast) single-molecule imaging.

## Results

### Reconstitution of synaptic vesicle/synapsin condensates
We first pursued a minimalist approach to reconstitute synaptic vesicle (SV)/synapsin condensates. Untagged, EGFP- and Halo7-tagged synapsin 1, hereafter referred to as synapsin 1, were purified using a mammalian expression system (Supplementary Fig. 1). Untagged synapsin 1 formed condensates similarly to tagged chimeric versions of the protein, as visualized by phase contrast microscopy. Native SVs were isolated from murine brains (Supplementary Fig. 2) and co-assembled with synapsin 1-driven condensates, as visualized upon the addition of FM 4–64 dyes to visualize SV lipid bilayer (Fig. 1a). SV/synapsin 1 condensates readily underwent fusion/fission, relaxing in round structures minimizing their surface tension (Fig. 1b). Note that doping the SV condensates with lipophilic FM 4–64 dye sometimes indicated a stronger signal at the periphery in several independent reconstitutions, presumably due to the penetration of dye or the local uneven distribution of SVs.

Next, we wanted to determine whether the presence of SVs affects the ability of synapsin 1 condensates to accumulate small solutes. To achieve this, we prepared the condensates and incubated them with fluorescently-tagged dextran[19,20]. This is further facilitated when the average mesh size of the pores within synapsin 1 condensates is larger than the hydrodynamic radius of dextran. Homogenous synapsin 1-only condensates accumulated 4.4 kDa (hydrodynamic radius ~2 nm) and 65.85 kDa dextran (hydrodynamic radius ~6 nm) while excluding 155 kDa dextran (hydrodynamic radius ~10 nm) (Supplementary Fig. 3). However, SV/synapsin 1 condensates could incorporate only 4.4 kDa while readily excluding 65.85 and 155 kDa dextran (Fig. 1c, d). To accommodate dextran, the loss of the enthalpy by the breakage of the synapsin-synapsin interactions needs to be compensated by the sum of the enthalpy of synapsin-dextran interaction and the increase in the mixing entropy. Thus, from our data, we can conclude that the presence of SVs (~40 nm diameter) increases the enthalpy of synapsin-dextran and decreases the mixing entropy thereby reducing the effective molecular weight cut-off from ~6 nm to ~2 nm (Fig. 1e). That SVs modulate the meshwork pore size of synapsin condensates was further confirmed by incubating the condensates with a mixture of fluorescently-tagged nanobody and primary/secondary antibody conjugates (Supplementary Fig. 4). While small nanobodies are readily enriched within condensates, the presence of SVs prevents larger primary/secondary antibody conjugates from penetrating inside the synapsin/SV condensates.

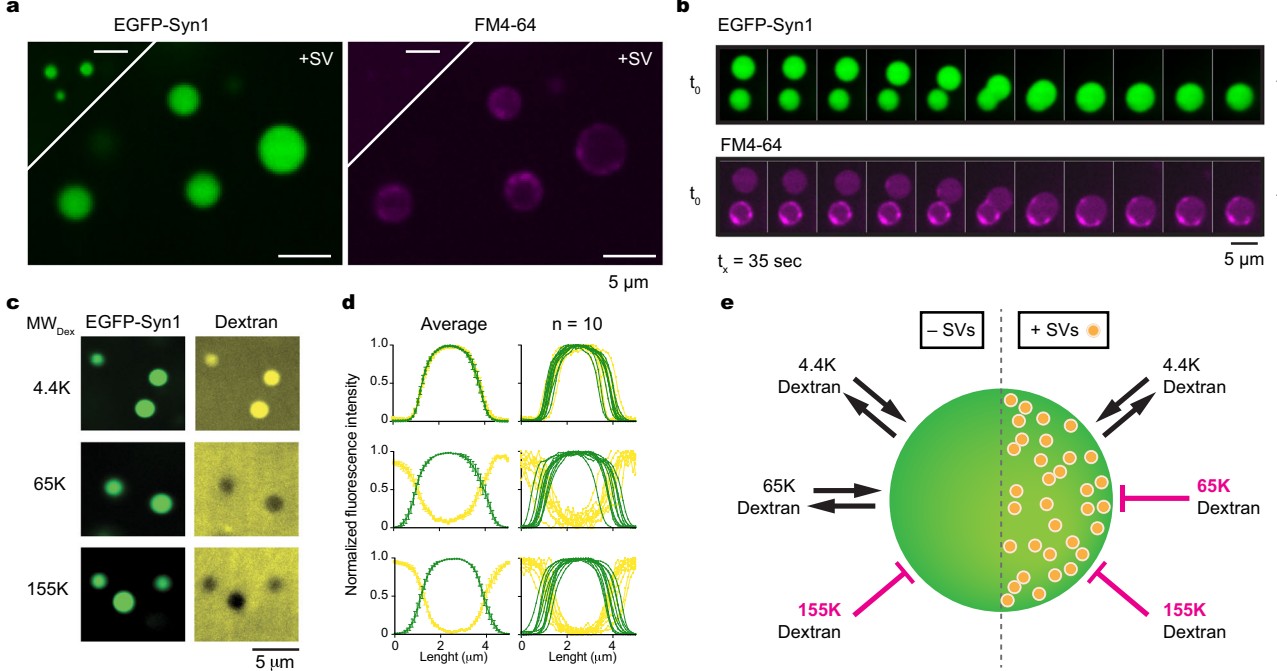

**Fig. 1 | Synapsin 1 and synaptic vesicles (SVs) form condensates. a** Confocal image of reconstituted EGFP-synapsin 1 (10 μM) in the absence (top) or presence (bottom) of natively isolated SVs (3 nM) doped with FM 4-64 dye to visualize the lipid bilayer. **b** Timelapse of synapsin 1/SV condensates fusing over a period of 35 s. **c** Confocal images of synapsin 1/SV condensates upon incubation with fluorescently-labeled dextran of 4.4 kDa (top), 65.85 kDa (middle), and 155 kDa (bottom) panels. **d** Line profiles indicating either enrichment or exclusion of fluorescently-labeled dextran of different molecular weights in synapsin 1/SV condensates. Left, an average ± SEM; right, exemplary line profiles. For each condition ten condensates of similar size from three independent reconstitutions are analyzed. **e** Scheme summarizing the effects of SVs on the average diameter of meshwork pore size in synapsin-driven condensates. Scale bars, 5 μm. Source data are provided as a Source Data file.

## Single-molecule detection of synapsin 1 in the condensates in vitro

Ensemble measurements of protein dynamics inside condensates, such as recovery after photobleaching, are quite useful for homogenous media but fail to distinguish short-range and long-range interactions and dynamics in non-homogeneous mixtures. To address this challenge, we turned to single-molecule tracking (SMT) coupled with total internal reflection fluorescence (TIRF) microscopy, which allowed us to obtain both high spatial (tens of nm) and temporal (ms) resolution of tracking. We reconstituted SV/synapsin 1 condensates that contained 1% of EGFP-tagged synapsin 1 (for visualizing the condensates), and 0.1% of Halo7(JF549)-synapsin 1 (for single-molecule tracking) on the cover-glass, and imaged the condensates with diameters between 2 - 4 μm on the cover-glass with a focus near their equatorial planes (Fig. 2a, b). The presence of 1% EGFP-tagged synapsin 1 hardly affected the diffusion properties of Halo7-synapsin 1 (Supplementary Fig. 5a).

We investigated the behaviors of Halo7-synapsin 1 at the level of single molecules projected onto the two-dimensional $x-y$ plane (for single-molecule localization precisions at 60-, 250-, and 1000-Hz observations, see Supplementary Fig. 5b, Supplementary Movie 1). First, we found that Halo7-synapsin 1 molecules in the condensate exchange with those in the bulk solution (Fig. 2b). For semi-quantitative analysis of the exchange rate, we identified synapsin 1 molecules that newly arrived at the condensate surface within the focal plane from the medium and measured the durations until they either left the condensate or became photobleached. Approximately 3 - 7% of Halo7-synapsin 1 molecules entered/disappeared from the field of view by diffusion in the z-direction, and we did not include these molecules in our quantitative analysis, since we concentrated our efforts on the analysis of the behaviors of Halo7-synapsin 1 molecules in the x−y plane, the percentages of these molecules were quite limited, and the z-direction behaviors would be the same as those in x−y planes in the core volume of the condensate, which was defined as regions >500 nm away from the surface. The distributions of dwell durations revealed the presence of two populations of synapsin 1 molecules with the shorter (≈0.76 s) and longer (≈27.1 s) dwell lifetimes in both condensates without and with SVs. As the shorter and longer dwell lifetimes were quite similar for the condensates, regardless of the SV presence, we use the average lifetimes to describe the shorter and longer fractions. These lifetimes were corrected for the photobleaching lifetime of JF549 dye (for statistical parameters, see Fig. 2c and Methods). The presence of SVs led to a 21% increase in the longer component (with a corresponding reduction in the shorter dwell lifetime component, Fig. 2c). These findings suggest that SVs interact with synapsin 1, consequently modulating its dynamics within the condensates.

## SVs globally suppress synapsin 1 diffusion in the condensates in vitro

Next, we examined the single-molecule movements of Halo7(JF549)-synapsin 1 in the core region of the condensates, projected onto the x−y plane (Fig. 2d). To avoid the strong edge effect caused by the projection, we selected synapsin 1 molecules located >500 nm away from the edge. Synapsin molecules exhibited rapid diffusion, both in the absence and presence of SVs. The analysis of single-molecule trajectories can be conveniently performed by plotting the mean-square displacement (MSD) against the elapsed time (Δt) (ensemble averaged over many observed molecules; Fig. 2e; Supplementary Fig. 5c; Methods). Such MSD-Δt plots could be simply analyzed by the equation, MSD = $4D_\alpha(\Delta t)^\alpha$, where $D_\alpha$ represents the diffusion coefficient near $\Delta t = 0$ and $\alpha$ represents the anomaly parameter. The anomaly parameter indicates the deviation of the observed diffusion from simple Brownian diffusion ($\alpha = 1$) and becomes smaller ($0 \leq \alpha < 1$) as the diffusion is more suppressed (subdiffusion).

We evaluated the single-molecule behaviors over a very broad range of time intervals, from a few milliseconds up to 3.2 s, to observe the possible long-range orders of synapsin 1 interaction and dynamics within the condensates and the effects of SVs, which are sparsely distributed in our SV-synapsin 1 reconstituted condensates, on synapsin 1 movements across various spatial ranges. To achieve this, we employed three different camera frame rates: 60, 250, and 1000 Hz (frame times of 16.7, 4, and 1 ms, respectively). Additionally, we generated 15-Hz and 30-Hz single-molecule trajectories (frame times of 67 and 33.3 ms, respectively) by re-sampling longer 60-Hz trajectories. Observations at various frame rates were necessary because, while ideal, 1000-Hz observations for up to 3.2 s (3200 frames) are impossible due to photobleaching. The MSD-Δt plots for 50 steps are presented for all the frequencies (Fig. 2e; note that the actual time in the x-axis is 50x the camera frame time and that the y-axis scales were varied accordingly), whereas the plot for 300 steps at 60-Hz (5 s) confirmed that the MSD-Δt plots based on 15- and 30-Hz re-sampled data were adequate (Fig. 2f).

The anomaly parameters ($\alpha$'s) obtained from the MSD-Δt plots in Fig. 2e are plotted against the total observation durations (the full x-axis scales in Fig. 2e) in Fig. 2g. Since all $\alpha$ values are <1, this result indicates that synapsin 1 diffusion within the condensate is suppressed from that expected for the simple-Browinan case, often called subdiffusion, even in the absence of SVs. This is likely to represent that the synapsin 1 condensates provide environments like a crowded and/or porous medium for the diffusion of synapsin 1 molecules that form the condensates, consistent with the dextran incorporation data (Fig. 1c−e). Smaller $\alpha$ values in the presence of SVs for the same observation duration indicate that the synapsin 1 diffusion is more suppressed in the condensates with SVs.

The plots in Fig. 2g demonstrate that the diffusion anomaly increases with longer observation durations for both condensates without and with SVs and that the reduction of $\alpha$ values slowly levels off in the duration range >0.83 s. To the best of our knowledge, such $\alpha$ dependences on the observation duration in protein condensates have not been reported thus far and are quite interesting. Such $\alpha$ dependences on the observed time duration are related to the diffusion suppression as a function of the diffused area. For instance, during a duration of 0.83 s, in the absence and presence of SVs, synapsin 1 molecules exhibited diffusion on the order of 2.7 and 1.7 x $10^{-2}$ μm² in MSD (60-Hz data in Fig. 2e) corresponding to ranges of (160- and 130-nm)², which are equivalent to ~16 and 13 molecules of synapsin 1 and to 4 and 3.3 SVs, respectively. Further reductions of $\alpha$-values for longer observation durations, 1.67 s and 3.3 s, for both condensates without and with SVs would indicate that there exist longer-range correlation lengths in the condensates than those covered here up to the observation time scale of 3.3 s (the space scales greater than a few tens of synapsin 1 molecules). This dependence of $\alpha$ on the observed time duration suggests that direct but multiple synapsin 1-synapsin 1 interactions propagate at least to a few tens of synapsin 1 molecules, which is consistent with the network mechanisms that protein condensates are considered to form[21]. Meanwhile, at all the temporal (and thus spatial) scales examined here, the SV-induced further suppression of synapsin 1 diffusion is evident.

Importantly, the SV-induced slowdown of synapsin 1 diffusion was also observed at a much shorter time scale of ≈3 ms (Fig. 2h). The diffusion coefficients for individual Halo7-synapsin 1 molecules were determined from their MSD-Δt plots in the range of 2 - 4 ms (without ensemble averaging) using 1000-Hz data ($D_{3ms}$), and their distributions are shown in Fig. 2h. In the presence of SVs in the condensates, both the mean and median $D_{3ms}$ values were smaller than those in the condensate without SVs. The distributions indicate a shift of the major peak toward smaller $D_{3ms}$ values, rather than the emergence of a secondary smaller peak for lower $D_{3ms}$ values, in the condensates containing SVs. This shift suggests that the presence of SVs broadly

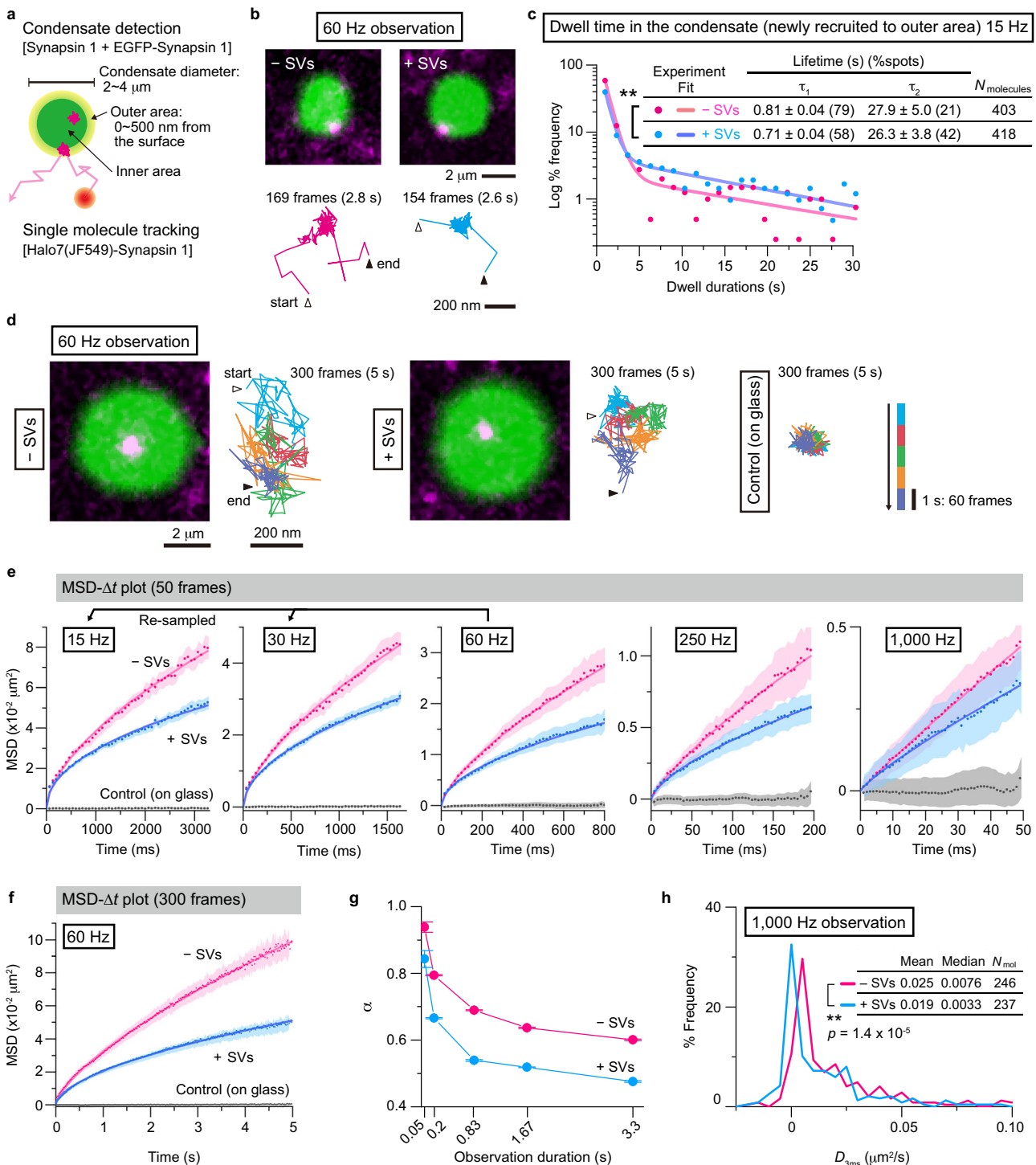

**Fig. 2 | Ultrafast SMT of synapsin 1 revealed the liquid properties of the SV/ synapsin condensates. a** Schematic drawing showing the design of synapsin 1 SMT experiments in the condensates in vitro. **b** Typical (among 314 and 356 Halo7(JF549)-synapsin 1 molecules in the experiments with and without SVs, respectively) fluorescence microscopy images of the synapsin 1 condensates marked by EGFP-synapsin 1 (green), recruiting a single Halo7(JF549)-synapsin 1 molecule (magenta), (top) and the representative trajectories of single Halo7(JF549)-synapsin 1 molecules (bottom). The images were recorded at 60 Hz in the absence (left) and presence (right) of SVs. **c** Distributions of the dwell durations of single Halo7-synapsin 1 molecules in the condensates; $p = 9.81 \times 10^{-14}$. **d** Representative (among 276 and 328 Halo7(JF549)-synapsin 1 molecules in the experiments with and without SVs, respectively) microscopy images including the Halo7-synapsin 1 molecules in the core regions, together with their trajectories in

the condensates core region. Control for immobile Halo7-synapsin 1 is also shown. The colors indicate the passage of time (see the time scale on the right). **e** The MSD-$\Delta t$ plots for Halo7-synapsin 1 molecules located in the inner volume (mean ± SEM). The full x-scales of all plots are 50 steps (50 frames). The number of observed molecules: 237, 237, 276, 317, and 246 (without SV, magenta); 307, 307, 328, 327, and 237 (with SV, cyan); and 242, 242, 476, 397, and 239 (on glass, gray), for frame rates of 15, 30, 60, 250, and 1000 Hz, respectively. For the offset value of the MSD, see Supplementary Fig. 5c. **f** The long-term MSD-$\Delta t$ plots (300 frames, 60 Hz; mean ± SEM). **g** The diffusion anomaly parameter ($\alpha$) depends on the total observation durations (mean ± SEM). **h** The distributions of the diffusion coefficients determined in the range of 2 - 4 ms, called $D_{3ms}$, evaluated from the trajectories obtained at 1000 Hz. $p = 1.41 \times 10^{-5}$; **$P < 0.01$, using Brunner-Munzel test for (**c**) and (**h**). Source data are provided as a Source Data file.

impacts the synapsin 1 diffusion in the condensate, rather than merely slowing the movement of synapsin 1 molecules bound on the surfaces of the SVs.

Considering that SVs occupy only a minor fraction of the volume in our reconstituted condensates (vesicle-to-protein ratio = 1:3000) mimicking a mere few percent of the synapse's total volume, the findings presented in Fig. 2g ($\alpha$ dependences on the observed time duration) and 2h ($D_{3ms}$ distributions) imply that SVs suppress the diffusion of synapsin 1 located on the SV surfaces through direct molecular interactions, and this slowing effect propagates quite far to exert a global effect, likely due to the synapsin 1 interactions that are indeed responsible for the condensate formation. Since the SVs are sequestered into condensates by the interaction with synapsin 1 molecules, it would be reasonable to conclude that this interaction reciprocally slows the dynamics of synapsin 1, although synapsin molecules continue to exchange rapidly between the condensates and the surrounding medium.

In addition, we examined the diffusion of the free JF549 dye in the synapsin 1 condensates without and with SVs in case that the unbound dye dominated the results described here (Supplementary Fig. 5d). From the MSD-$\Delta t$ plots, the diffusion of free dye is clearly much faster than those incubated with Halo7-synapsin 1. However, interestingly, the free dye also exhibited more suppressed diffusion (smaller anomaly parameter) in the condensates with SVs. This might be due to the small size of the dye, which allows its transient local accumulation in the condensates.

## Single-molecule imaging of synapsin 1 and SVs in live neurons

We then aimed to scrutinize the motility of synapsin and SVs in living neurons. We co-transfected primary hippocampal neurons with Halo7-synapsin 1 (synapsin 1) and synaptophysin-mEos3.2 (synaptophysin), a bona-fide SV marker, and synapsin 1 was labeled with fluorogenic dye Janelia Fluor 635 (JF635). Single-molecule localization in live neurons showed a clear accumulation of synapsin 1 in synaptic boutons and its colocalization with SVs (Fig. 3a, Supplementary Movie 2). This stereotypical localization allowed us to separately assess the motility of synapsin 1 and SVs within and between boutons by performing two-color SMT experiments along the axon (Fig. 3b). In contrast to synaptophysin-mEos3.2 which is almost exclusively confined inside boutons (Fig. 3b, c), synapsin 1 is present along the entire axons (Fig. 3b, d). Although enriched at the synaptic boutons, synaptophysin remains mobile with a mean diffusion coefficient of 0.049 ± 0.005 $\mu m^2 s^{-1}$ and the anomaly parameter $\alpha$ of 0.65 ± 0.014 (Fig. 3f,g). For an integral membrane protein, the surprisingly high apparent diffusion of synaptophysin may result from its oligomerization in the membrane of SVs[22], the pronounced membrane fluidity of SV bilayer[23], and the motion of SV as a whole. Given that the spatial resolution achieved in the live tracking experiments (~15 nm) is in the same range as the average radius of SVs (~20 nm[24]), it is not possible to distinguish between the movement of individual synaptophysin molecules in the SV membrane and the overall movement of SVs. Rather, we use the motility of synaptophysin here as a proxy for the relative motility of SVs. Interestingly, we identified two populations of synapsin 1 (Fig. 3e, h, i): the first is confined at synaptic boutons (geometric mean diffusion = 0.051 $\mu m^2 s^{-1}$ and $\alpha$ = 0.65 ± 0.024), while the second is much less confined (geometric mean diffusion = 0.18 $\mu m^2 s^{-1}$ and $\alpha$ = 0.84 ± -0.044) and exhibits high diffusivity between boutons (Fig. 3e, h, i) resembling the motility of mEos3.2 tag alone (Supplementary Fig. 7). Calculated from the number of trajectories, we observed a ~3-fold enrichment of synapsin 1 molecules inside versus outside boutons. This pattern of synapsin 1 diffusion remains independent of expression levels, as indicated in our measurements of synapsin 1 on one- and 2 days post-transfection (Supplementary Fig. 6). To delineate synapsin 1 confinement driven by LLPS from diffusion occurring outside the condensates in the boutons and in between the

boutons due to non-specific crowding, we compared the motion of mEos3.2 alone with that of mEos3.2-synapsin 1 (Supplementary Fig. 7). Indeed, mEos3.2-synapsin 1 was more confined and less mobile inside boutons than soluble mEos3.2 despite their molecular weight being within the same order of magnitude. This indicates that the confinement of synapsin 1 in boutons is a specific effect independent of molecular crowding.

To avoid selection bias, we repeated the entire experimental pipeline and filtered the acquired trajectories not by location (e.g., inside or outside synaptic boutons) but rather by the level of confinement with 0–0.75 for confined motion and >1.25 for directed motion (Supplementary Fig. 8). Importantly, we identify a similar pattern with two population of synapsin 1 molecules with a largely confined and mobile population inside synaptic boutons. The trajectories of synapsin 1 that followed directed motion were only appearing between neighboring boutons.

Additionally, we generated synapsin 1 condensates in non-neuronal cells, where synapsin is not endogenously expressed. Functionally, these synapsin 1 condensates actively recruit SVs and SV-like vesicles in the reconstituted systems[16]. Specifically, we co-transfected CV-1 cells with plasmids encoding synapsin 1 and a concatemer of SH3 domains of intersectin. Intersectin is a cytoplasmic membrane-associated protein, which contains multiple SH3-domains essential for the generation of liquid-liquid phase separated condensates[25]. Indeed, we observed the formation of condensates in CV-1 cells reminiscent to synapsin 1 condensates in primary neurons. In these ectopically-formed condensates synapsin 1 was accumulated and confined irrespectively whether the condensates were induced by co-expression of mEos3.2-synapsin 1 with the SH3 concatemer, Grb2, or synaptophysin (Supplementary Fig. 9), strongly suggesting that synapsin 1 has the capacity to form condensates irrespectively of SVs both in vitro, in neurons and in ectopic systems. However, the presence of other components in the system and local differences in ion and protein composition may regulate the viscoelastic properties of these condensates[20].

## IDR of synapsin 1 is crucial for SV motility and confinement in vivo

To complement our analysis in synapses in cultured neurons of wild-type mice, we turned to a murine model where all three synapsin genes are deleted (i.e., synapsin triple knock-out mice, SYN-TKO). Here, the number of SVs and their packing in the synapses were shown to be significantly lower both in neurons in culture and in brain slices[7,8]. We investigated the motility and level of confinement of SVs in SYN-TKO neurons (Fig. 4 a–c, Supplementary Fig. 10). Indeed, SVs were significantly more mobile and less confined in SYN-TKO than in wild-type neurons (Fig. 4 e, f). Importantly, we fully rescued both the confinement and motility of SVs in SYN-TKO neurons expressing full-length mEos3.2-synapsin 1 (Fig. 4 d–f). Synapsin 1 diffusion is independent of the type of tag used for tracking, mEos3.2 or Halo7 (Supplementary Fig. 7). In contrast to the clear effect of synapsin 1 on SV motion in SYN-TKO neurons, there was no effect of co-expression of synaptophysin on synapsin 1 motility or confinement in wild-type neurons (Supplementary Fig. 7). This is congruent with the genetic analysis that the number of totally available SVs remains unaltered by the expression levels of synaptophysin[26].

Synapsin 1 contains a large intrinsically disordered region (IDR)[27]. IDRs are sequences of amino acids that do not fold into any stable secondary or tertiary structure. IDR of synapsin 1 (a.a., 416-705, syn1-IDR) was shown to be necessary and sufficient for triggering LLPS in vitro[7,10]. Thus, we suspected that syn1-IDR is sufficient for the unique diffusion signature of SVs. To test this, we purified syn1-IDR (Supplementary Fig. 1) and reconstituted it with natively isolated SVs. Indeed, the condensates of syn1-IDR were able to sequester SVs similar to the full-length proteins (Supplementary Fig. 11). We further expressed syn1-

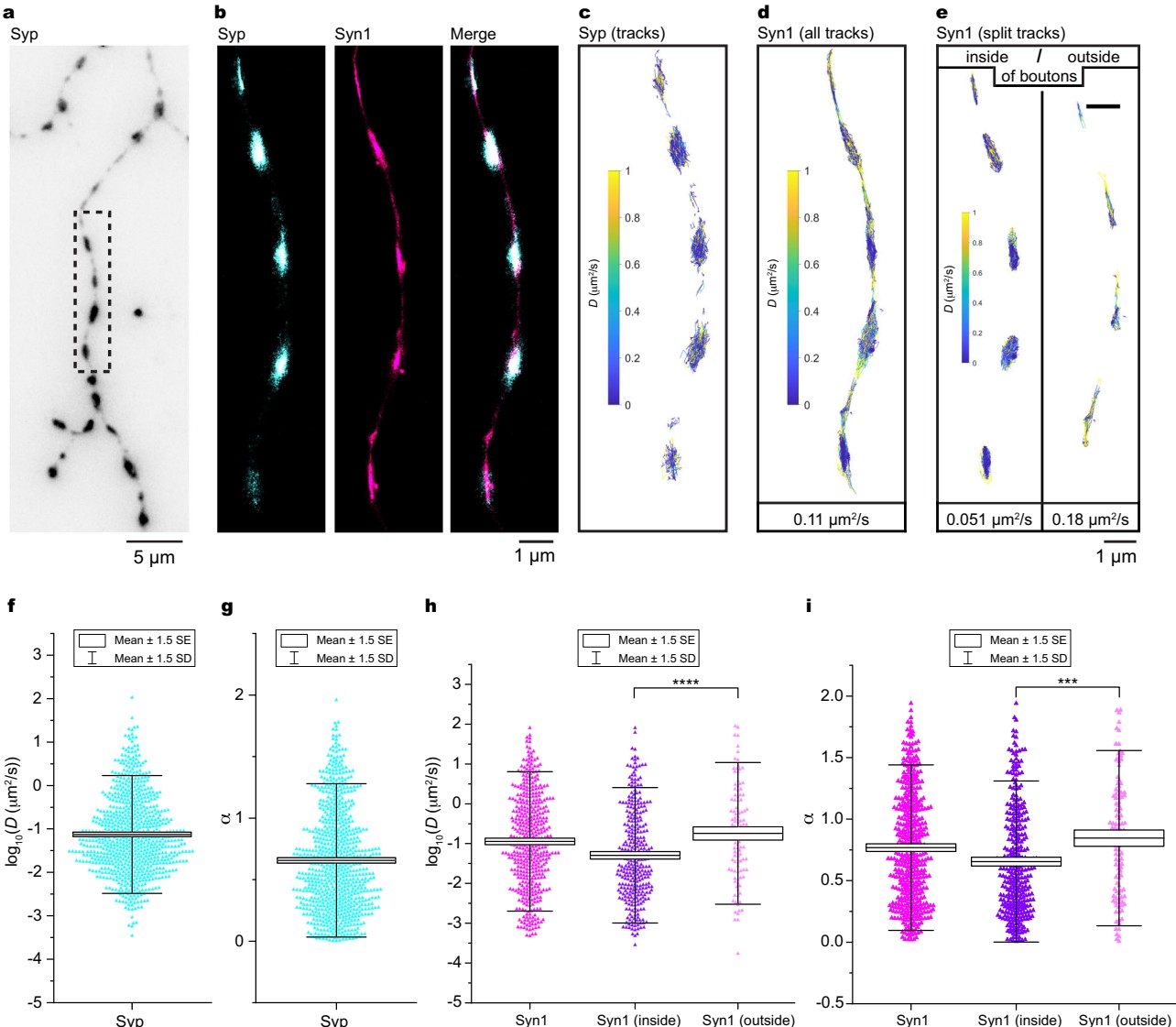

**Fig. 3 | Synapsin 1 and synaptophysin are confined at synaptic boutons while maintaining their high motility. a** Single-molecule localization image of a neuron expressing synaptophysin-mEos3.2 (Syp). Mouse hippocampal neurons (14 days in culture) were transfected with synaptophysin-mEos3.2 (Syp) and Halo7-synapsin 1 (Syn1). **b** Single-molecule localization reconstruction of proteins localized within dashed box in (**a**). Dual-color single-molecule tracking of Syp or Syn1 coupled to JF635 were performed at 100 Hz for 50 s. **c** Map of all tracks ($n = 775$) of Syp within dashed box in (**a**) color-coded for diffusion coefficient. Note that Syp is almost exclusively confined at synaptic boutons. **d** Map of Syn1 within dashed box in (**a**) color-coded for diffusion coefficients indicating all tracks ($n = 464$). **e** Syn1 tracks inside synaptic boutons (left, $n = 337$) and between boutons (right, $n = 116$), indicating 2.9x enrichment of synapsin 1 inside boutons. Values indicate the geometric mean diffusion coefficient for each panel. Boxplots (mean ± 1.5 × SD (whiskers) and 1.5 × SEM (box)) showing diffusion coefficients (**f**) and the coefficient of confinement (α) (**g**) for all tracks of Syp shown in (**c**). Boxplots (mean ± 1.5 × SD (whiskers) and 1.5 × SEM (box)) showing diffusion coefficients (**h**) and α (**i**) for all and grouped tracks of Syn1 shown in (**d**, **e**). Significance was tested using Mann–Whitney test; asterisks, statistical significance; n.s., not significant. Note that Syn1 is significantly slowed down (**h**, $p = 1.4 \times 10^{-5}$,***) and confined (**i**, $p = 1.2 \times 10^{-4}$,****) in synaptic boutons. Source data are provided as a Source Data file.

IDR in SYN-TKO neurons. 3D stimulated depletion emission (STED) microscopy indicates that syn1-IDR is targeted to the synaptic bouton similarly to synapsin 1 full-length (Fig. 4a). The syn1-IDR not only rescued the phenotype that is, the clustering of SVs in boutons (Supplementary Fig. 12) but also led to a further reduction in motility of SVs as compared to either wild-type neuron or a rescue with a full-length synapsin 1 (Fig. 4d–f). This clearly indicates that syn1-IDR is sufficient to generate condensates and to recruit SVs to them in vivo. In addition to changes in diffusive behavior, we observed a decrease in accumulation of SVs in boutons in SYN-TKO neurons as compared to wild-type neurons, as reported previously[7,8]. This lack of accumulation was fully rescued by the expression of full-length synapsin 1. Overexpression of just syn1-IDR in SYN-TKO neurons was already sufficient to trigger an increase of SVs accumulation in synaptic boutons (Fig. 4b, c).

This raised an important question whether syn1-IDR would be sufficient to rescue the clustering of SVs in vivo. To assess this question, we turned to functional analysis of SV exocytosis by using pH-sensitive probe (i.e., the luminal domain of synaptophysin that contains pHluorin) and performed the rescue experiments in SYN-TKO neurons (Supplementary Fig. 14, Supplementary Movie 3). Excitingly, both IDR and full length of synapsin 1 rescued the SV release in SYN-TKO neurons (Fig. 4g). This is particularly interesting since synapsins 2 and 3 contain shorter IDR and no IDR, respectively[27]. In wild-type synapses, synapsins can heterodimerize through their highly-conserved, dimerization motif[28]. Yet syn1-IDR solely was sufficient to recapitulate clustering and secretion of SVs. As these are hippocampal neurons in culture and the chemical stimulation with high (90 mM) potassium chloride is comparatively strong to common physiological

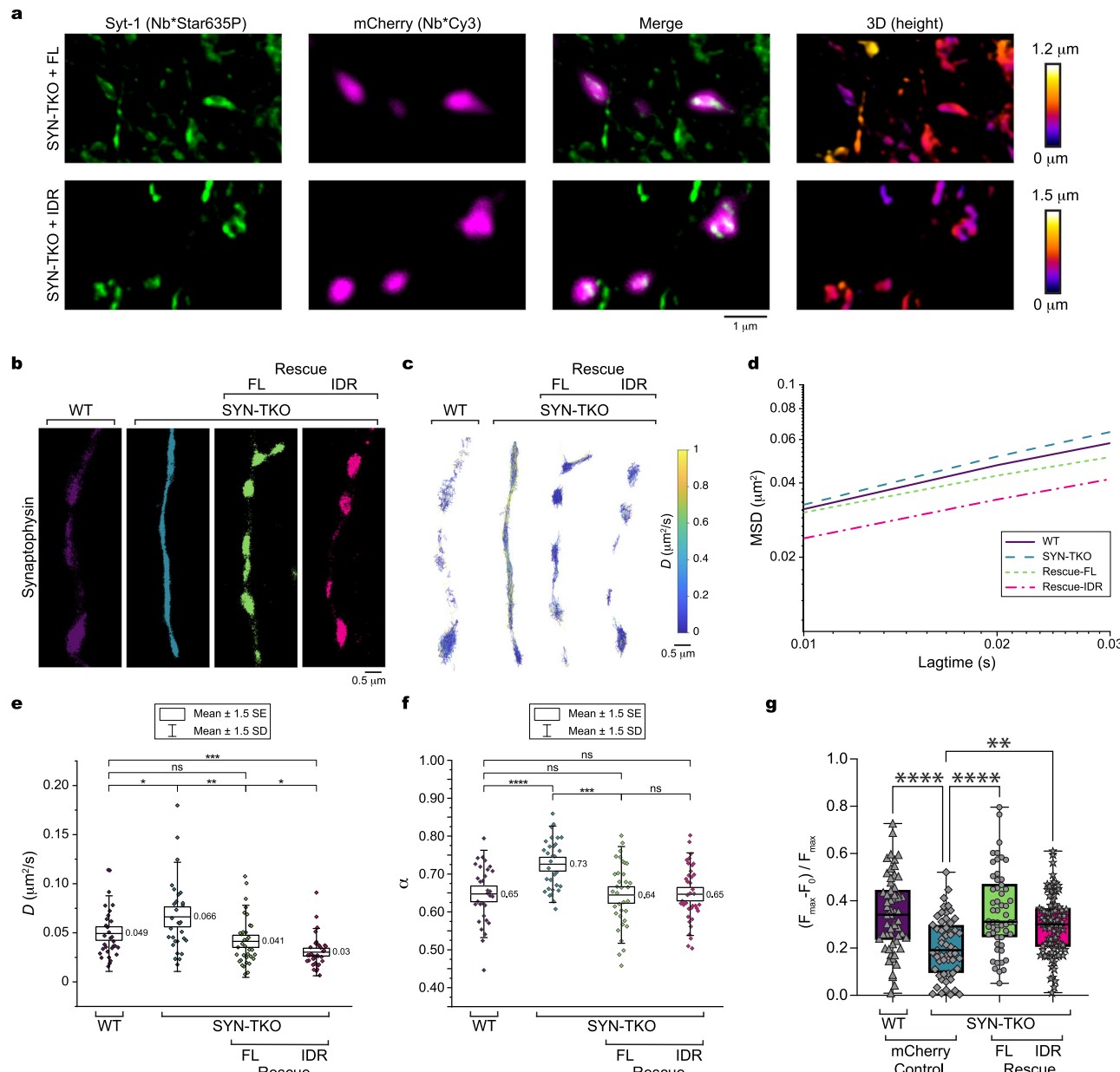

**Fig. 4 | Synapsin determines the confinement and diffusion rates of SVs. a** 3D-STED imaging of synapsin triple knockout (SYN-TKO) neurons expressing mCherry-tagged full-length (top) or the intrinsically disordered region (IDR; bottom) of synapsin 1. Primary hippocampal neurons were fixed and stained with fluorescently-tagged nanobodies against synaptotagmin−1, a bona fide synaptic vesicle protein (Nb*Star635P, STED channel), and against mCherry (Nb*Cy3, confocal channel). **b** Single-molecule localization reconstructions from tracking experiments of synaptophysin-mEos3.2 (Syp) performed at 100 Hz for 50 s. Mouse hippocampal neurons (14 days in culture) from wild-type or SYN-TKO animals were transfected with Syp. Rescue experiments were done by co-transfecting SYN-TKO neurons with Syp and either full-length synapsin 1 (Rescue FL) or IDR of synapsin 1 (Rescue IDR). **c** Images of all Syp tracks for each condition. **d** Mean log(MSD)-log(lag)-plot for all tracks per condition. Note that the rescue with IDR shows even greater reduction of motility than the FL rescue. **e** Boxplots (mean ± 1.5 x SD

(whiskers) and 1.5 x SEM (box)) showing the geometric mean diffusion coefficients per tracking experiment of at least four independent neuronal preparations and a total of 77,675 analyzed tracks. **f** Same as in (*e*) but showing the mean coefficient of confinement (α) per tracking experiment (WT, *n* = 31; SYN-TKO, *n* = 31; Rescue-FL, *n* = 35; Rescue-IDR, *n* = 39). In *e* and *f*, the significance was tested using Mann-Whitney test; asterisks indicate significance: *$p < 0.1$; **$p < 0.01$; ***$p < 0.0001$; ****$p < 0.00001$; ns, not significant. Note that SVs are significantly faster and more dispersed in the absence of synapsins. **g** SV release measured as an increase in fluorescence of pH-sensitive EGFP variant tagged to the luminal region of synaptophysin upon chemical stimulation with KCl (90 mM). Wild-type or SYN-TKO neurons from three independent neuronal preparations were co-transfected with only a soluble red fluorophore, synapsin FL or the synapsin IDR. Significance was tested using one-way ANOVA test; asterisks, significance: **$p < 0.01$; ***$p < 0.0001$. Original data are provided as a Source Data file.

stimuli, the specific protein-protein interactions or distinct synapsin isoforms might be central to regulate the secretion across different frequencies[29].

Importantly, to avoid bias by the type analysis chosen, we analyzed the data generated from single-molecule tracking experiments in three different ways: by treating each tracking experiment as a data point (Fig. 4c, e, f), by treating each track as a data point (Supplementary Fig. 13), or by calculating MSD-lagtime-curves (Fig. 4 d). Regardless of the analysis type, the observed trends of synapsin 1 and SV diffusion remained the same. Interestingly, taking the amount of synapsin 1 (ectopically expressed in the absence or presence of endogenous synapsin) as a proxy for the total protein concentration,

the data suggests that the increased levels of synapsin 1 further confine and reduce the motion of SVs as reminiscent to the in vitro phase diagrams[19]. Taken together, we conclude that synapsin 1 is a master regulator of SV motility and confinement at synaptic boutons in living neurons.

## Discussion

Through the quantitative characterization of synapsin 1 and SV dynamics, the findings reported here strongly support that condensation of synapsins and SVs suffices to induce SV microdomain formation without the need for a (complex) 'molecular fence' that would restrict SVs within a bouton. This does not exclude an additional role for protein–protein interactions between either integral membrane proteins or soluble proteins. In fact, these seem essential for exchanging the SVs between neighboring synapses[17] or re-incorporating newly endocytosed vesicles[30,31].

The enrichment of SVs in synaptic boutons by LLPS has clear consequences for neuronal transmission. For example, our in vivo measurements show that SVs maintain their motility in synaptic boutons despite them being confined in a synapsin-dependent manner (Fig. 4). SV release can occur at a wide range of different stimulation frequencies[32]: during low-activity, docked SVs and SVs at the interface would be recruited for fusion, whereas, during prolonged activity, the synapsin 1-driven condensate will provide a reservoir of SVs[29,33]. Initially, it was proposed that synapsins act as tethers by crosslinking the vesicles together[34]. This implies that synapsins play a role in physically connecting the vesicles, thereby maintaining their clustered arrangement. The high local concentration of synapsins suggests them being more than mere crosslinkers of SVs[13,14]. We and the other showed that synapsins can form a liquid phase, effectively ensnaring the synaptic vesicles within it[7,10,18]. In this scenario, synapsins act as a cohesive medium, keeping the vesicles together in a clustered state while allowing for their high motility[4,35,36]. It is also plausible that a combination of both mechanisms occurs, as they are not necessarily mutually exclusive[12].

The diffusive behavior of SVs is completely perturbed in SYN-TKO neurons, being more mobile and less confined. Surprisingly, syn1-IDR was sufficient to restore both the diffusion pattern and the accumulation of SVs at boutons. This may be due to the last patch of amino acids within the IDR, so-called Domain E, which are well-conserved in *a*-type isoforms of all three synapsins[27,37]. We speculate that both the phase separation and the specific interactions provided by these last amino acids may allow for the coupling of SV clustering to neurotransmitter release[38].

Finally, both in vitro (0.019 $\mu m^2/s$) and in living neurons (0.051 $\mu m^2/s$), synapsins maintain their fast motility within these condensates despite their spatial confinement. This allows synapsin molecules to be rapidly phosphorylated by kinases upon the neuronal activity at the interface of condensates without the need for enzymes to fully penetrate into condensates. Thus, the size of meshwork coupled to the internal dynamics of proteins within a condensate provides an additional layer of specificity and allows for SV condensates to act as buffers of proteins and enzymes[39,40].

Collectively, our findings that SV/synapsin condensation is sufficient for vesicle sequestering and dynamics strongly support that such interactions constitute a mechanism for the formation of organelle microdomains in the cytosol by a small set of proteins. We suggest that a cooperation of mechanisms that create mesoscale domains of vesicles and proteins plays a role in other types of cellular assemblies[41,42], although the precise players implicated in such assemblies and the resulting diffusion patterns will be context-specific.

## Methods

The research in this study complies with all relevant ethical regulations. All animal experiments were approved by the Institutional Animal Welfare Committees of the State of Berlin, Germany and Charité University Clinic (Berlin, DE), as well as State of Lower Saxony, Germany and Max Planck Institute for Multidisciplinary Sciences (Göttingen, DE).

### Cloning

*His14-SUMO_Eu1-mmSynapsin1a* plasmid was created by exchanging the mScarlet-I tag of a *mScarlet-I-mmSynapsin 1a* expression vector (NM_013680.4) with the His14-SUMO_Eu1 tag cassette[43]. The His14-SUMO_Eu1 cassette was amplified from pAV0279 (*His14-SUMO_Eu1-MBP-3xFLAG*) with primers #DML0331 & #DML0332 and inserted into *mScarlet-I-mmSynapsin 1a* vector using *Age*I and *Bgl*II restriction sites.

*6His-HaloTag-rnSynapsin 1* plasmid was created by exchanging the EGFP tag with Halo7Tag in an EGFP-rnSynapsin 1a expression vector[7]. The HaloTag was amplified from LYN-Halo7 with primers #DML0195 & #DML0196 and inserted into *EGFP-rnSynapsin 1a* vector using *Age*I and *Bgl*II restriction sites. For Ni-NTA affinity purification, a 6xHis tag was introduced by Annealed Oligo Cloning at the N-terminus of the HaloTag using *Nhe*I and *Age*I (#DML 0438 & 0439).

*SYPH-pHluorin* plasmid was created by Gibson assembly resulting in the expression of murine synaptophysin M7-A240 (NM_009305.2) with an inserted pH-sensitive GFP cassette (pHluorin[44]) between T190 and C191 under control of a CMV promoter and SV terminator (similar as in[45]). All constructs were verified by sequencing.

For primers and detailed sequences, see Supplementary Tables 1–2. pAV0286 for bacterial expression of SENP_EuB protease and pAV0279 as the template for His14-SUMO_Eu1 were from the lab of Dirk Görlich (Addgene plasmid #149333; http://n2t.net/addgene:149333; RRID:Addgene_149333 and Addgene plasmid #149688; http://n2t.net/addgene:149688; RRID:Addgene_149688)[43].

### Protein purification

*His14-SUMO_Eu1-mmSynapsin1a* was expressed in Expi293F™cells (Thermo Fisher Scientific) for 3 days following enhancement. Cells were lysed in buffer that contained 25 mM Tris-HCl (pH 7.4), 300 mM NaCl, 0.5 mM TCEP (buffer A) supplemented with EDTA-free Roche Complete protease inhibitors, 15 mM imidazole, 10 μg/μL DNaseI and 1 mM MgCl$_2$. All purification steps were carried out at 4 °C. Debris was removed by centrifugation for 30 min at 20,000xg. For batch affinity purification, the soluble supernatant was incubated with complete His-Tag Purification resin (Roche) on a rocking platform for 1 h. Washing steps (buffer A) and elution (buffer A with 400 mM imidazole) were performed by gravity flow in a polyprep column (Biorad). Elution fractions were concentrated using a 30 K MWCO protein concentrator (Pierce) and subjected to size exclusion chromatography (Superdex™ 200 Increase 10/300, GE Healthcare, ÄKTA pure 25 M) in buffer A. Fractions containing *His14-SUMO_Eu1-mmSynapsin1a* were combined and digested overnight with SENP_EuB SUMO protease for tag-cleavage (protease:protein ratio of 1:20). For tag removal, the reaction was supplemented with 15 mM imidazole and passed 3 times over preequilibrated complete His-Tag Purification resin (buffer A). Tag-free mmSynapsin1a protein in the flow-through was concentrated with a 30 K MWCO protein concentrator and subjected to dialysis against 25 mM Tris-HCl (pH 7.4), 150 mM NaCl, 0.5 mM TCEP. Proteins were snap-frozen in liquid nitrogen and stored at -80 °C until further use.

SENP_EuB SUMO protease was recombinantly purified as described previously[43]. Specifically, SENP_EuB SUMO protease was expressed in bacteria (Rosetta (DE3) cells with plasmid pAV0286). The protein expression main culture was grown to an OD$_{600}$ of 1.0 in LB medium containing kanamycin at 37 °C. Protein expression was induced with 0.3 mM IPTG final concentration. After overnight protein expression at 18 °C culture was supplemented with 5 mM EDTA. Cell pellets were harvested and subjected to lysis (French-press) in buffer containing 45 mM Tris-HCl (pH 7.5), 290 mM NaCl, 15 mM imidazole, 4.5 mM MgCl$_2$, 10 mM DTT (buffer B) supplemented with 10 μg/mL DNaseI and

a spatula tip of lysozyme. The lysate was cleared by centrifugation for 1 h at 50,000xg (JA25.50) at 4 °C. All purification steps were carried out at 4 °C. For batch affinity purification, the cleared soluble supernatant was incubated with complete His-Tag Purification resin (Roche) on a rocking platform for 1 h. Washing steps (45 mM Tris-HCl (pH 7.5), 500 mM NaCl, 15 mM imidazole, 4.5 mM MgCl$_2$, 10 mM DTT) and elution (buffer B with 400 mM imidazole) were performed by gravity flow in a polyprep column (Biorad). SENP_EuB-containing fractions were subjected to size exclusion chromatography (HiLoad™ 16/600 Superdex™ 200 pg, Cytiva, ÄKTA pure 25 M) in 45 mM Tris-HCl (pH 7.4), 290 mM NaCl, 4.5 mM MgCl$_2$, 2 mM DTT. SENP_EuB fractions were combined and supplemented with 2.5% (v/v) glycerol for storage at -80 °C until use.

*EGFP-synapsin 1, EGFP-synapsin 1 IDR (416-705)* and *Halo7-synapsin 1* were expressed in Expi293F™cells (Thermo Fisher Scientific) for 3 days following enhancement. Cells were lysed in a buffer that contained 25 mM Tris-HCl (pH 7.4), 300 mM NaCl, 0.5 mM TCEP (buffer A) supplemented with EDTA-free Roche Complete protease inhibitors, 25 mM imidazole, 10 μg/mL DNaseI and 1 mM MgCl$_2$. All purification steps were carried out at 4 °C. Debris was removed by centrifugation for 1 h at 20,000xg. For affinity purification, soluble supernatant was applied on a Ni-NTA column (HisTrap™HP, Cytiva, ÄKTA pure 25 M) for binding. After a wash step (buffer A with 40 mM imidazole) and elution (buffer A with 400 mM imidazole), proteins were concentrated and subjected to size exclusion chromatography (Superdex™ 200 Increase 10/300, GE Healthcare, ÄKTA pure 25 M) in 25 mM Tris-HCl (pH 7.4), 150 mM NaCl, 0.5 mM TCEP. Proteins were snap-frozen in liquid nitrogen and stored at -80 °C until further use.

## In vitro Halo7-synapsin 1 labeling

For fluorescent labeling of Halo7-synapsin 1, 250 μg purified protein (in 25 mM Tris-HCl (pH7.4), 150 mM NaCl, 0.5 mM TCEP) was incubated with 0.7 μg Janelia Fluor® HaloTag® ligand (JF549, Promega GA111A; stock: 0.13 μg/μL in DMSO) for 1 h at room temperature. The reaction was applied to PD-10 columns (Sephadex G-25, Cytiva) equilibrated with 25 mM Tris-HCl (pH7.4), 150 mM NaCl, 0.5 mM TCEP. Elution fractions containing labeled Halo7(JF549)-synapsin 1 protein fraction were concentrated using a 30 K MWCO protein concentrator (Pierce) and snap-frozen in liquid nitrogen for storage at -80 °C until use.

## Synaptic vesicle preparation

Synaptic vesicles (SVs) were isolated according to previous publications[24,46,47]. Briefly, 20 rat brains (isolated from adult male rats) were homogenized in ice-cold sucrose buffer (320 mM sucrose, 4 mM HEPES-KOH, pH 7.4 supplemented with 0.2 mM phenylmethylsulfonylfluoride and 1 mg/ml pepstatin A). Cellular debris was removed by centrifugation (10 min at 900 g$_{Av}$, 4 °C) and the resulting supernatant was further centrifuged for 10 min at 12,000 g$_{Av}$, 4 °C. The pellet containing synaptosome was washed once by carefully resuspending it in sucrose buffer and further centrifuged for 15 min at 14500 g$_{Av}$, 4 °C. Synaptosomes were lysed by hypo-osmotic shock and free, released SVs were obtained after centrifugation of the lysate for 20 min at 20,000 g$_{Av}$, 4 °C. The supernatant containing the SVs was further ultracentrifuged for 2 h at 230,000 g$_{Av}$, yielding a crude synaptic vesicle pellet. SVs were purified by resuspending the pellet in 40 mM sucrose followed by centrifugation for 3 h at 110,880 g$_{Av}$ on a continuous sucrose density gradient (50–800 mM sucrose). SVs were collected from the gradient and subjected to size-exclusion chromatography on controlled pore glass beads (300 nm diameter), equilibrated in glycine buffer (300 mM glycine, 5 mM HEPES, pH 7.40, adjusted using KOH), to separate synaptic vesicles from residual larger membrane contaminants. SVs were pelleted by centrifugation for 2 hr at 230,000 g$_{Av}$ and resuspended in sucrose buffer by homogenization before being aliquoted into single-use fractions and snap frozen in liquid nitrogen. The weight/volume of SV proteins is related to the copy/number of SVs as described in (7).

## Reconstitutions

For all reconstitutions, we co-incubated 10 μM synapsin 1 with 1% (w/v) PEG 8000. When synaptic vesicles were included in the reconstitutions, we first added FM4-64 dye to synapsin 1 and then triggered the condensate formation by the addition of 1% (w/v) PEG 8000. Synaptic vesicles (3 nM final concentration) were loaded with a low-binding pipette tips. The assay scheme is shown in Supplementary Data Fig. 2. Reconstitution reactions were performed in a buffer containing 25 mM Tris-HCl (pH 7.4), 150 mM NaCl, 0.5 mM TCEP.

**Dextran-based pore size determination.** All TMR-dextrans were purchased from Sigma (T1037-50MG, T1162-100MG, T1287-50MG). For pore size analysis of synapsin 1 phase, condensates were formed by incubating 10 μM EGFP-synapsin 1 with 1% (w/v) PEG 8000 on a glass bottom dish (Cellvis D35-20-1.5-N). TMR-dextrans were added to the condensates when they reached an average size of 5 μm in diameter (0.04 mg/mL TMR-dextran final concentration). The participation of dextrans in synapsin 1 phases was analysed by confocal imaging (Eclipse Ti Nikon Spinning Disk Confocal CSU-X, 2 EM-CCD cameras (AndorR iXon 888-U3 ultra EM-CCD), Andor Revolution SD System (CSU-X), objectives PL APO 60x/1.4 NA, oil immersion lens. Excitation wavelengths were: 488-nm for EGFP; 561-nm for TMR).

**Nanobody/antibody-based pore size determination.** For the analysis of pore size, synapsin 1 condensates were formed by incubating 10 μM synapsin 1 (9 μM non-tagged synapsin 1 + 1 μM Halo7-tagged (JF549 labeled) synapsin 1) with 1% (w/v) PEG 8000 in the presence or absence of 3 nM synaptic vesicles. The reaction was incubated on ice for 5 min before placing on a glass bottom dish. After 10 min a nanobody/antibody premix consisting of FluoTag-X2*AF488 nanobody (NanoTag Biotechnologies, N1202-AF488, anti-Mouse IgG kappa light chain sdAb), anti-LAMP1 antibody (Sigma, L1418, produced in rabbit), and goat-anti-rabbit IgG-Cy5 antibody (Cytiva, RPN999) was added to the synapsin or synapsin/SV condensates on the glass bottom dish (final nano-/antibody concentrations: 0.01 mg/mL, 0.1 mg/mL and 0.01 mg/mL respectively). To test for different partitioning of the nano-/antibodies images were taken after 10 min of incubation (Excitation wavelengths: 488-nm for FluoTag-X2*AF488 nanobody; 561-nm for Halo7-tagged (JF549 labeled) synapsin 1, 647-nm for goat-anti-rabbit IgG-Cy5 antibody).

**Sequestering of synaptic vesicles by the full-length or the intrinsically disordered region of synapsin 1.** 10 μM EGFP-synapsin 1 or 10 μM EGFP-synapsin 1 IDR (amino acids: 416-705) was incubated with 5% (w/v) PEG 8000 in the presence or absence of 3 nM synaptic vesicles on glass bottom dishes. SVs were preincubated for 5 min on ice with either an SV-specific AbberiorStar635P-labeled anti-Synaptotagmin nanobody (NanoTag Biotechnologies, #N2302-Ab635P-L, FluoTag-X2 anti-Synaptotagmin 1) or lipophilic FM4-64 dye (AAT Bioquest, #21487, MM 4–64 dye) before adding to synapsin condensates with a diameter in the range of 3–4 μm (final concentrations: 125 nM and 1.65 μM respectively). The presence of synaptic vesicles in the condensates was visualized by SDC microscopy (Excitation wavelength: 488-nm for EGFP, 561-nm for FM4-64 and 647-nm for Ab635P-labeled anti-Synaptotagmin 1).

## Primary hippocampal neurons

Hippocampal neurons were prepared from P0/1 mice of WT (C57BL/6) or synapsin triple-knockout (SYN-TKO: B6; 129-Syn2tm1Pggd Syn3tm1Pggd Syn1tm1Pggd/Mmjax)[8] background. Brains were manually dissected and hippocampi were collected in cold Hank's balanced salt solution (HBSS, Gibco) containing 10 mM Hepes buffer (Gibco),

1 mM Pyruvic Acid (Gibco), 0.5% penicillin/streptomycin and 5.8 mM Magnesium Chloride. After dissection, hippocampi were enzymatically digested with Papaine (Sigma) in HBSS for 20 min at 37 °C. Papaine was removed with repeated HBSS washing, and plating medium was added (Gibco Neurobasal Medium A [NB-A], supplemented with 5% FBS, 1% B27, 1x Glutamax, and 1% penicillin/streptomycin). Final cell suspension was obtained through mechanical dissociation with a P1000 pipette. Cells were seeded on glass coverslips coated with 0.1 mg/mL poly-L-lysine (PLL; Sigma). Neurons were maintained at 37°C and 5% $CO_2$ in Neuronal Media (NB-A supplemented with 1% B27, 1% Glutamax, and 0.5% penicillin/streptomycin).

Neurons were transfected by calcium phosphate transfection (adapted from[48,49]). In brief, a coverslip with neurons was transferred to a fresh petri dish containing 1 ml growth medium supplemented with 4 mM kynurenic acid (Sigma K3375, 20 mM stock solution in NB-A, freshly prepared). Transfection mix contained 2 µg of each construct in 1x TE (10 mM Tris-HCl (pH 7.3), 1 mM EDTA) supplemented with $CaCl_2$ to a final concentration of 250 mM (stock: 10 mM HEPES (pH7.2), 2.5 M $CaCl_2$). The mix was added dropwise to 2xHEBS (42 mM HEPES (pH 7.2) 274 mM NaCl, 10 mM KCl, 1.4 mM $Na_2HPO_4$, 10 mM glucose) with slow vortexing between each addition and incubated for 20 min at room temperature. Transfection mix was added to neurons and dishes were incubated at 37 °C, 5% $CO_2$ for 1.5 h. Medium was replaced by 1 mL of NB-A with 4 mM kynurenic acid supplemented with 2.5 mM HCl. Wash was performed at 37 °C, 5% $CO_2$ for 15 min. After transfection coverslip was placed back in the original culture dish with own conditioned medium.

## Quantifying SV release in primary hippocampal neurons

Images in two channels were taken under the spinning-disc confocal microscope using the wavelengths of 488 nm for the SYPH-pHluorin and of 561 nm for the mCherry/mCherry-synapsin 1. The coverslips containing neurons were mounted in pre-warmed low-KCl Tyrode solution (150 mM NaCl; 4 mM KCl; 2 mM $CaCl_2$; 2 mM $MgCl_2$; 10 mM HEPES; 10 mM glucose; pH at 7.4 using NaOH) before imaging them under the microscope. After an initial z-stack ($F_0$), high-KCl pre-warmed Tyrode solution (154 mM KCl; 2 mM $CaCl_2$; 2 mM $MgCl_2$; 10 mM HEPES; 10 mM glucose; pH at 7.4 using NaOH) was added to the coverslip to obtain a final concentration of 64 mM NaCl and 90 mM KCl; an established approach for stimulating neurons in culture coupled to live-cell imaging recordings[49]. Immediately after adding the correspondent volume, another z-stack was taken at the same position ($F_{max}$). Then, $NH_4Cl$ was added to the coverslip to a final concentration of 50 mM, and a final z-stack was acquired. Images were analysed in Fiji, taking the maximum intensity of the selected ROIs for both the low-KCl and high-KCl images. Calculations were performed (($F_{max} − F_0)/F_{max}$), averaged and plotted.

## Single-molecule tracking

**High-speed single-molecule tracking in neurons.** All samples were imaged using a Vutara 352 super-resolution microscope (Bruker) equipped with a Hamamatsu ORCA Flash4.0 sCMOS camera for super-resolution imaging and a 60x oil immersion TIRF objective with a numerical aperture of 1.49 (Olympus). Immersion Oil (#1261, Cargille) was used for the live-cell tracing in CV-1 cells or primary mouse hippocampal neurons, 14 days in culture. All experiments were done in live cell imaging solution (Thermo Fisher, 14291DJ) at 37 °C. All data acquired in neurons was collected from at least 3 biological replicates from independent preparations. Cells transfected with mEOS3.2-contructs were briefly illuminated with a low-intensity 405 laser (0.0022 kW*$cm^{−2}$). Afterwards, mEOS3.2-constructs were tracked at 100 Hz for 5000 frames at a laser power of 0.49 kW*$cm^{−2}$ with a 561 nm laser under HiLo-Illumination. Cells transfected with Halo7-constructs were incubated with 100-500 pM JF635-Halo7 for 5 min. Afterwards, cells were washed with live cell

imaging solution. Then, Halo7-labeled constructs were tracked at 100 Hz for 5000 frames at a laser power of 0.57 kW*$cm^{−2}$ with a 641 laser under HiLo-Illumination. For dual-color measurements, tracking of Halo7- and mEOS3.2- constructs was performed sequentially as described above.

**Data processing of single-molecule tracking data.** Acquired raw data were localized using SRX (Bruker). Localizations were estimated by fitting single emitters to a 3D experimentally determined point spread function (PSF) under optimization of maximum likelihood. The maximum number of localization iterations performed before a given non-converging localization was discarded, was set to 40. PSFs were interpolated using the B-spline method[50]. For single molecule localization reconstructions, locations were rendered according to their Thomson accuracy[50]. For single-molecule tracking analysis, localizations were exported using SRX (Bruker) and tracked in 3D using the FIJI Mosaic tracker plugin (Linkrange: 1, Displacement: 5)[51]. Diffusion coefficients and alpha exponents were calculated in Mosaic[51]. All tracks were filtered according to their α (2 > α > 0). For further analysis the mean alpha and geometric mean diffusion coefficient of all tracks per measurement were calculated.

We analyzed the single-molecule tracks either according to their localization (i.e., regions in boutons vs. the regions outside boutons) or the filtering of the tracks according to coefficient of confinement (0 ≤ α ≥ 0.75) vs super-diffuse (1.25 ≤ α ≥ 2). In the latter approach, the tracks of intermediate confinement including the Brownian motion (0.75 ≤ α ≥ 1.25) are filtered out completely. Statistical tests for significance (Mann-Whitney) were performed in Origin (OriginLab). Alternatively, all tracks of all measurement per conditions were pooled to calculate log(MSD):log(Δ$t$) curves using a self-written MATLAB code (MathWorks).

**In vitro ultrafast SMT of synapsin 1.** Synapsin condensates were reconstituted, as described above, in the final solution containing 150 mM NaCl and 0.5 mM TCEP buffered with 25 mM Tris−HCl, pH 7.4. Final synapsin 1 concentrations were 10 µM, 100 nM, and 10 nM for non-tagged, GFP-tagged, and Halo7-tagged (JF549 labeled) molecules, respectively. After the addition of a final concentration of 1% PEG 8000 (Sigma Aldrich, P5413), the mixed solution was placed on the glass-bottom slide (Ibidi, 81507). All observations were finished within 15 min after PEG addition.

Fluorescently labeled molecules were excited by oblique-angle illumination using an objective-lens-type TIRF microscope, which was constructed on an Olympus inverted microscope (IX-83) equipped with a 100x 1.49 NA objective lens (Olympus, UApoN 100x TIRF). The incident excitation laser intensities at the sample plane were 8.2 µW/µm² for the 488-nm line for GFP (Coherent, Sapphire 488-300 CW), 1.4, 5.7, and 12.7 µW/µm² for the 561-nm line for JF549 (MPB Communications, Fiber 560-20000) for the observation frame rates of 60, 250, and 1000 Hz, respectively (note 1 µW/µm² = 0.1 kW/cm²). The simultaneously-obtained two-color fluorescence images were separated into two detection arms of the microscope by a dichromatic mirror (Chroma, ZT561rdc-UF3), filtered through band-pass filters of ET525/50 m or ET600/50 m (Chroma), and the images were projected onto the high-speed camera system (Photron, mini AX200-II) described previously[52,53]. The final magnification was 250x (~80 nm/pixel; square pixels). Under these observation conditions, localization precisions were determined with dyes fixed on the glass: 22.1 and 22.5 nm for x- and y-coordinates at a frame rate of 60 Hz, 29.9 (x) and 30.0 (y) nm at 250 Hz, and 33.9 (x) and 33.0 (y) nm at 1000 Hz, respectively (for the data, see Supplementary Fig. 5b).

The shapes of the condensates were determined by averaging the EGFP-synapsin 1 images recorded at 60 Hz over 100 frames. Individual Halo7-synapsin 1 spots in the image were identified and tracked by using an in-house computer program, as described previously[54,55]. The

superimposition of images in different colors obtained by two separate cameras was conducted as described previously[56].

**Determination of the dwell lifetime of Halo7-synapsin 1 (JF549 labeled) in the condensate in vitro.** For evaluating the diffusion properties of Halo7-synapsin 1 in the condensate, we analyzed its diffusion projected onto the 2D xy plane, assuming that synapsin 1 movements in the z-direction is the same as those in x and y-directions in the core region of the condensate. Furthermore, we characterized the general molecular behaviors before quantitatively examine the diffusion of Halo7-synapsin 1 within the condensate: we measured the dwell lifetime of Halo7-synapsin 1 (JF549 labeled) in the condensate by using a time-lapse observation (15 Hz; every 66.7 ms, but the camera exposure time and laser illumination duration was 16.7 ms for each observation), which was useful for lowering the effect of photobleaching. Under these observation conditions, the photobleaching lifetime of JF549 was 54.4 s (13.6 s at 60 Hz observation).

As the arrival of a new molecule, we only included the molecules that arrived at the edges of the condensates from the outside of the condensate in/near the focal plane (within ≈ ±200 nm from the exact focal plane because we only selected the clear spots) and excluded those that came into clear focus by diffusion in the z-direction within the condensate. This was because we intended to analyze the diffusion projected onto the 2D xy plane. Less than ~5% of Halo7-synapsin 1 molecules entered the view within the condensate this way.

Halo7-synapsin 1 molecules "departed" from the view field in the condensates by (1) the departures from the edges of the condensates into the medium that occurred in/near the focal plane (within ≈ ±400 nm from the exact focal plane), (2) by photobleaching while they were still clearly visible in the focal plane within the condensates, and (3) by diffusing out of the focal plane in the z-direction (> ±400 nm from the exact focal plane) in the condensates. The molecules that disappeared by the process (3) were not included in the analysis by the same reason that the molecules that diffused into the view field by the z-direction diffusion were not considered for newly arrived molecules. Only ≈7% and ≈3% of the total Halo7-synapsin 1 molecules we observed disappeared in this manner in the condensates without and with SVs, respectively. The small numbers of arriving and departing molecules due to diffusion in the z-direction indicate that this method of analysis would not be skewed due to the possible omission of rapidly diffusing molecules.

The distribution of the Halo7-synapsin 1 dwell durations (15 Hz time-lapse observations with 16.7 ms exposure time) was fitted by the sum of two exponential functions, providing two dwell lifetimes. Observed dwell lifetimes ($\tau_{\text{observed}}$) were corrected for the photobleaching lifetime of the probe[57], as follows,

$$\frac{1}{\tau_{\text{true}}} = \frac{1}{\tau_{\text{observed}}} - \frac{1}{\tau_{\text{bleach}}}, \tag{1}$$

where $\tau_{\text{true}}$ and $\tau_{\text{bleach}}$ are the true dwell lifetime and photobleaching lifetime.

**Analysis of the MSD-Δt plots obtained from ultrafast SMT of Halo7-synapsin 1 in the condensates in vitro.** MSD-Δt plots were calculated for up to 50 and 300 frames (out of the initial 100 and 350-frame trajectories; Fig. 2e, f, respectively) and averaged over 200 ~ 500 trajectories. The 15- and 30-Hz trajectories were generated by re-sampling the 60-Hz data.

The MSD-Δt plots were evaluated based on an anomalous diffusion model using the following equation,

$$\text{MSD} = \langle r^2(t) \rangle = 4D_\alpha(t)^\alpha + C, \tag{2}$$

where $r$, $t$, $D_\alpha$, $\alpha$, and $C$ are displacement, time, diffusion coefficient, the anomaly parameter, and offset due to the localization error, respectively. The MSD-Δt plots after and before the offset subtraction are shown in Fig. 2e and Supplementary Fig. 5c, respectively. In the MSD-Δt plots for Halo7-synapsin 1 bound to the glass shown in Fig. 2e, the mean of the MSD values at all the Δt points were subtracted.

**Sample preparation for 3D STED microscopy.** 3D STED microscopy were performed on 4 different sets of samples- wild type, Syn TKO, full-length synapsin 1 transfected and synapsin 1 IDR transfected neuronal cultures (grown on coverslips), which were fixed using 4% PFA. Transfected samples (full-length synapsin 1 or IDR) were labeled with two fluorophores. The first was introduced to identify the synapses of interest. For this purpose, transfected cells were stained with a Cy3-conjugated mCherry nanobody. The second fluorophore was introduced by immunostaining SVs using a STAR 635P-conjugated synaptotagmin 1 nanobody, which was used to acquire STED images for all sets of samples. Sample preparation for STED microscopy was done as previously described[58]. Specifically, fixed cells were blocked and permeabilized with 2.5% BSA, 2.5% NGS, 2.5% NDS and 0.1% Triton X-100 in PBS for 30 min at room temperature. Synaptotagmin 1 (Nanotag, #N2302-Ab635P-L) and m-Cherry (Nanotag, #N0404-SC3-L) nanobodies were diluted to a final concentration of 25 nM in blocking buffer before applying to the coverslips. The cells were incubated with nanobody solution for 1 h followed by washing with permeabilization buffer (5 buffer exchanges for 1 h) and two quick washes with PBS. Finally, the coverslips were mounted onto the slides using antifade mounting media (Thermo Fisher Scientific, P36980).

**Acquisition and analysis of 3D-STED data.** An Abberior Expert line setup (Abberior Instruments) equipped with an IX83 inverted microscope (Olympus) was employed to generate confocal and STED images. A 100X oil immersion objective (UPLXAPO, 1.45 NA; Olympus) was used to focus the excitation light into the sample plane. Star635P was excited with a 640 nm pulsed excitation laser (set to 5% of max. power, 1.5 μW, frequency of 80 MHz), while a 561 nm excitation laser (27% of max. power, 1.5 μW; 80 MHz) was used for Cy3. Fluorescence signals were detected using an avalanche photodiode (APD) that has dedicated preset ranges for the two dyes (650-720 nm for Star635P and 605-625 nm for Cy3). Stimulated depletion for StarS635P was obtained by a 775 nm depletion laser (set to 6.4% of max. power, 5 mW). 3D STED images were acquired with a step size of 50 nm in z, pixel size of 20 nm, a dwell time of 10 μs per pixel and a line accumulation of 3.

Data were analyzed using Matlab (the Mathworks, Inc.). The synapses of interest were identified by applying an empirically-derived threshold on the Cy3 images. The Star635P signals within these images were filtered using a bandpass filter, and the locations and intensities of all discernable spots were determined and analyzed statistically, as described in the respective figures.

**Statistics and reproducibility**
The number of biological and technical replicates and the number of analyzed molecules are indicated in the figure legends and Source Data. Curve fittings were performed using Origin 2017 (OriginLab). Statistical tests were performed using R-studio (with the lawstat package) and Prism 9 (GraphPad). Differences were considered statistically significant for $p$-values < 0.05. The representative images in Figs. 1a and 2d come from at least three independent reconstitutions using the material (i.e., recombinant proteins and isolated SVs) from at least three independent preparations. The STED images in Fig. 4a were repeated from three independent neuronal preparations from wild-type and SYN-TKO animals.

**Reporting summary**

Further information on research design is available in the Nature Portfolio Reporting Summary linked to this article.

## Data availability

Source data are provided with this paper. All data generated or analyzed for this study are available within the paper and its associated Supplementary Information/Source Data file. Source data are provided with this paper.

## Code availability

Superimpositions of image sequences obtained in two colors and tracking single-molecules for in-vitro ultrafast observation data were perfomed using C + +-based computer programs, produced in-house[52] and based on the well-established approaches[59]. Source codes for this have been integrated into a large, complex software package. While the software package cannot be extracted in a useful way, the entire software is available from A.K. upon reasonable request, who will provide guidance on how to use it, as its manual and comments are written in Japanese. Tracking in live cells and neurons was done using Mosaic tracker plugin[51]. The code to generate figures with corresponding data is available in Source Data file.

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

## Acknowledgements

We thank Advanced Medical Bioimaging Core Facility at Charité for the support; Kristina Jevdokimenko (UMG Göttingen) for the assistance with STED. The work is supported by the start-up funds from DZNE, as well as the grants from the European Research Council MemLessInterface–101078172 to DM; and the German Research Foundation SFB 1286/B10 and MI 2104 to DM; and SFB 1286/A03 to SOR. CH is supported by a fellowship of the Innovative Minds Program of the German Dementia Association; RC is supported by a fellowship from the Alexander von Humboldt Foundation. Schematics in Supplementary Data Fig. 2 and 14 are created with BioRender.com.

## Author contributions
D.M. conceptualized the study. C.H. did all the reconstitution experiments and analyzed the data. T.A.T. and A.K. performed in-vitro single-molecule tracking and analyzed the data. J.R. and H.E. performed single-molecule tracking experiments in neurons and analyzed the data. R.C., A.H.S., and S.O.R. performed 3D STED microscopy. G.G. contributed to single-molecule microscopy. C.H., A.C., A.A.K., and M.G. performed partitioning measurements; G.A.P. and F.T. performed experiments in neurons. C.H., J.R., T.A.T., A.K., H.E., and D.M. wrote the paper. All the authors read and approved the final version of the paper.

## Funding

## Competing interests
The authors declare no competing interests.
