## [Peer review file · Nature Communications]

REVIEWER COMMENTS

Reviewer #1 (Remarks to the Author):

Synapsin plays critical roles in synaptic vesicle (SV) maintenance and mobilization. Accumulating evidence in the past few years suggest that synapsin, via liquid-liquid phase separation, can organize the reserve pool SVs. The senior author of this manuscript made this original discovery and has been playing a leading role in this direction of research. However, whether and, if yes, how synapsin and SV may reciprocally affect the dynamics of each other in vitro and in synapses are not known. In this manuscript, using single-molecule tracking (SMT) the authors discovered that formation of condensate slows down motions of Synapsin both in in vitro reconstituted synapsin/SV condensates and in cultured hippocampal neurons. They further discovered that the intrinsically disordered sequence of synapsin 1 is sufficient to restore the dynamic properties and functions of SV in neurons derived from synapsin triple knock-out animals. The study provides a single molecular level view of the motion properties of synapsin in the synapsin/SV condensates and adds additional evidence showing critical roles of synapsin in clustering SVs. The manuscript is a strong candidate for NC. However, a number of technical issues will need to be addressed to make sure the derived conclusions are rigorous.

1. Regarding to the dextran-based mesh size estimation (Figure 1): this finding is rather surprising that the reconstituted synapsin/SV condensate has a mesh pore size less than 6 nm (i.e. excluding 65 kDa dextran). The synapsin only condensate has a mesh pore size <10 nm. It is possible that the molecular properties of dextrans may be a reason for their exclusions by the synapsin condensates. The authors will need to use another method (e.g. using inert proteins with different hydrodynamic radii) to double check the approximate pore sizes of synapsin condensates.

2. Ext. Data. Fig. 2, the workflow of sample loading does not match with the description in the method (supp. p5 lines 151-). When were SVs added? Before or after synapsin 1 condensation? The authors proposed that stronger FM 4-64 signal at the periphery was due to penetration of the dye (p4, lines 88-). A control experiment with FM4-64 dye on synapsin only condensates is needed. An alternative explanation by the authors is due to local uneven distribution of SVs, probably corresponding to the "outer area" as defined in Figure 2. This is an interesting point, as the boundaries in condensed droplets of phase separated systems are very interesting topic. The question is how the MSD curve and mobility parameters of synapsin 1 in the outer area (or boundary area) would look like? Would these parameters be different from synapsins in the inner area? According to the experimental pipeline described in the manuscript, SV labelling with FM4-64 was simply achieved by mixing the dye with SVs followed by immediate imaging. How do authors control the labelling of SVs by the dye (e.g., how the experimental conditions were determined for optimal dye labelling)?

3. Figure 2, the authors used the dwell time to describe the exchange properties but without any method description. What are the criteria for defining molecules “newly arrived” at the condensate and “left” the condensate? Also, despite the photobleaching lifetime of the fluorescent molecules, molecules diffuse out of focal plane will contribute to the signal loss during tracking the single molecule localizations. Can the authors add more details on this and discuss the influence of above-mentioned situations?

4. Figure 2c, the authors showed two typical tracks of synapsin 1 with or without SVs. However, the cumulative displacement as well as the maximum MSD of the molecules are small, which indicate the molecules are confined or oscillate within a small region. Such motions do not seem to match properties of molecules that undergo more or less random walk in the condensed phase. Did they observe longer range displacements in their images? One would expect that at least for certain trajectories the range explored by synapsin 1 may cover several hundreds of nm or even at μm level. Can the authors show some example trajectories of molecules entering or leaving the condensed phase (as these trajectories will likely be much longer)? The resolution (or precision/detection uncertainty) of the fluorescent signal is proportional to the square root of photon number and thus proportional to the square root of imaging frequency. How was the resolution (precision) distribution and the mean resolution of the tracks in different imaging frequency were defined? If the resolution is about the same level of what the authors claimed, the motions of molecules may be masked by the detection uncertainty of the microscope. For example, at 1000 Hz, the authors observed even in the absence of SV, synapsin 1 only experienced a ~ 67 nm oscillation, which is almost the same level of the estimated signal resolution under the experiment condition. Thus, the molecule’s true diffusion could be much smaller than what the authors have observed, making the comparisons between different conditions not meaningful.

5. Figure 2d & f, the authors used anomalous diffusion model to analyse the MSD. Oddly, the diffusion anomaly parameter α decreased with imaging frequency (p6 line 149). The diffusion anomaly parameter should be independent of the sampling frequency. The fact that the anomaly parameter changes with detection frequency further indicates that the observed diffusion may be masked by the detection uncertainty. Another odd result is the control MSD curve of “on glass” group are with extremely low MSDs and sometimes even with negative values (especially in 1000 Hz). Since mean square displacement should always be a non-negative value, can the author elaborate why the negative MSD values were presented? As mentioned before, the detection uncertainty of the signal will create artificial displacements, and the level is about $2 \times 20 \times 20 \text{ nm}^2 \sim 0.001 \mu\text{m}^2$ even with a resolution (precision) as high as 20 nm. If only consider the contribution of photon numbers, the resolution in 1000 Hz should be at least 4 times to that in 60 Hz, the contribution of MSD would be $\sim 0.01 \mu\text{m}^2$, which is much higher than what the authors showed in Figure 2d. Can the authors explain why the control MSD values are so low?

6. Figure 2 (in vitro reconstitution) concluded that the mobility of synapsin 1 was decreased in the presence of SV, and the authors speculated that “the interaction with synapsin 1 reciprocally slows the dynamics of synapsin by prolonging its dwell lifetimes in the condensates” (p6, lines 162-). But Ext. Data.

Figs. 6 and 8 (cultured hippocampal neurons), the authors concluded that “synapsin confinement and diffusion rates are independent of SV” (supp. P12, line 309; p8, line 205). How should we understand the role of SV on synapsin 1 mobility?

7. Ext. Data. Fig. 8, the authors ectopically co-expressed synapsin 1 and intersectin in CV-1 cells and resulted in condensates formation and confinement of synapsin 1. The estimated D was $\sim 0.05 \mu\text{m}^2/\text{s}$, which is comparable to D of synapsin 1 in hippocampal neurons (Figure 3). Presumably CV-1 cells do not contain SV like structures, does this mean the confinement of synapsin 1 is an intrinsic property of the molecules within a condensate (i.e., independent of SVs as the authors have stated)? Additionally, both D values obtained in neurons and in CV-1 cells are much higher than those derived from the in vitro reconstituted system (Figure 2, in the absence of SV, $\sim 0.02 \mu\text{m}^2/\text{s}$ for 1000 Hz, and ~ 0.008 for 60 Hz), how to reconcile these results obtained from different systems?

8. P6, lines 159: the authors mentioned “fraction of the volume”. Do authors have a way to estimate the volume fraction of synapsin 1 and SV? When the authors used “3 nM” to describe the SV concentration, how did they determine the concentration of SV? Each SV is associated with hundreds of proteins (PMID: 17110340)¹, then 3 nM SV may correspond to a few μM proteins. EM graphs clearly showed that majority space of a bouton (or the reserved pool SV area) is occupied by the SVs. Thus, the volume fraction of SVs should be considered together with synapsin 1.

9. Figure 3 and 4: the authors found that synaptophysin (as the mark for SVs) has a similar diffusion property compared to synapsin 1. Does this mean a comparable mobility of SV and synapsin 1? This is counterintuitive, as one would expect that SVs move much slower than synapsins. Furthermore, the authors treat synaptophysin movement same as SV movement. Synaptophysin may undergoes redistribution within an SV and move along with the SV macroscopically. How to differentiate these two types of motions of synaptophysin? It will be critical to carefully analyse trajectories of synaptophysin. One would expect that both step sizes and overall MSD of synaptophysin should be $< 50 \text{ nm}$, which is about the size of an SV. Again, with such short step size, the imaging resolution becomes to be a critical factor.

10. Figure 4: it's surprising that syn1-IDR was sufficient to rescue SV distribution and SV release. Here the IDR in this manuscript covers synapsin 1 D and E domains, and the domain D contributes to the majority of the length and governs the phase separation property (ref 10)². It is surprising that the major SV binding domains (C and A) are not involved in synapsin 1's function. If the authors wish to claim that synapsin 1-D and E domain are sufficient to restore SV defects, additional experimental evidence, including EM images of recovered SV distributions and in vitro reconstitution of synapsin 1-DE recruitment of SVs are required. In addition, an early synapsin-TKO study suggested that synapsin 2, but NOT synapsin 1, can rescue synaptic depression in glutamatergic synapses (PMID: 18945891)³, how to reconcile these results with the current finding in the manuscript?

11. Also in Figure 4: many studies including some from the senior author have demonstrated that synapsin TKO led to drastic SV loss, particularly the reserve pool. Considering different pools of SVs have different mobilities (PMID: 22442064, 20643088)^{4, 5}, how the authors evaluate their measured SV mobilities in boutons of WT and synapsin-TKO synapses?

Refs:

1. Takamori S, et al. Molecular anatomy of a trafficking organelle. *Cell* 127, 831-846 (2006).
2. Pechstein A, et al. Vesicle Clustering in a Living Synapse Depends on a Synapsin Region that Mediates Phase Separation. *Cell reports* 30, 2594-2602 e2593 (2020).
3. Gitler D, Cheng Q, Greengard P, Augustine GJ. Synapsin IIa controls the reserve pool of glutamatergic synaptic vesicles. *The Journal of neuroscience : the official journal of the Society for Neuroscience* 28, 10835-10843 (2008).
4. Kamin D, et al. High- and low-mobility stages in the synaptic vesicle cycle. *Biophys J* 99, 675-684 (2010).
5. Orenbuch A, et al. Synapsin selectively controls the mobility of resting pool vesicles at hippocampal terminals. *The Journal of neuroscience : the official journal of the Society for Neuroscience* 32, 3969-3980 (2012).

Reviewer #2 (Remarks to the Author):

Hoffmann and co-workers present a microscopy-based study on highlighting how the protein synapsin controls sequestering and dynamics of extracellular vesicles through condensation (or liquid-phase separation). They show this for in vitro and cell measurements as well as link it to signalling events. Many controls are included, which makes this story very convincing. These are important insights with high impact. This certainly deserves publication.

One general issue though: The authors are very fixed on LLPS. However, modification of EVs by synapsin has also been indicated through other processes such as aggregation of synapsin. The transition (or distinguishing) between aggregation and LLPS is any way very smooth. Can the authors at all comment on this, maybe the data could be described through a more unifying instead of contradictory model?

I have a few small comments:

- Page 4, line 82: How large are the condensates?
- General to the diffusion data (whether in vitro or in cells): Did the authors check for free dye, unbound to the Halo-Tag? Maybe show one example with another dye or highlight the diffusion for the free dye (as they did once)? Also, how did the authors include noise in the MSD data analysis?
- Page 5, lines 133ff: This sentence is not clear – why “even” and what “resulting confinement”?
- Page 5, line 140: How did the authors determine the radius of gyration or the diameter of the SVs?
- Page 6, lines 152ff: I did not quite get how the authors got to this conclusion.
- Page 6, line 159: Give numbers for the molar ratios here.
- Page 6, lines 161ff: Where is the proof for this statement?
- General: Why and how did the authors use PALM? This does decrease the temporal resolution for single-molecule tracking it seems. Why did they not just use a low enough labelling concentration?
- Page 7, line 175: Quantify the enrichment!
- Page 7, lines 177: I got a little bit confused here. Which population is slowed-down, which one is confined to certain regions and which one diffuses between boutons? My confusion mainly stems from the fact that confinement is used both for slow-down and confinement into regions. Also, always state numbers for alpha.
- Page 9, lines 245ff: Maybe give comparing numbers for diffusion coefficients here?
- Page 10, line 280: What is meant by fast here? Difficult to get a feeling for it after all the discussed confinement.

POINT-BY-POINT RESPONSE TO:

SYNAPSIN CONDENSATION CONTROLS SYNAPTIC VESICLE SEQUESTERING AND DYNAMICS

Reviewer #1 (Remarks to the Author):

Synapsin plays critical roles in synaptic vesicle (SV) maintenance and mobilization. Accumulating evidence in the past few years suggest that synapsin, via liquid-liquid phase separation, can organize the reserve pool SVs. The senior author of this manuscript made this original discovery and has been playing a leading role in this direction of research. However, whether and, if yes, how synapsin and SV may reciprocally affect the dynamics of each other in vitro and in synapses are not known. In this manuscript, using single-molecule tracking (SMT) the authors discovered that formation of condensate slows down motions of Synapsin both in in vitro reconstituted synapsin/SV condensates and in cultured hippocampal neurons. They further discovered that the intrinsically disordered sequence of synapsin 1 is sufficient to restore the dynamic properties and functions of SV in neurons derived from synapsin triple knock-out animals. The study provides a single molecular level view of the motion properties of synapsin in the synapsin/SV condensates and adds additional evidence showing critical roles of synapsin in clustering SVs. The manuscript is a strong candidate for NC. However, a number of technical issues will need to be addressed to make sure the derived conclusions are rigorous.

We thank the Reviewer for the positive assessment of our work and the constructive suggestions throughout the revision points. We performed all suggested experiments, which confirmed our initial hypotheses, and we re-wrote the text, according to the suggestions of the Reviewer.

1. Regarding to the dextran-based mesh size estimation (Figure 1): this finding is rather surprising that the reconstituted synapsin/SV condensate has a mesh pore size less than 6 nm (i.e. excluding 65 kDa dextran). The synapsin only condensate has a mesh pore size <10 nm. It is possible that the molecular properties of dextrans may be a reason for their exclusions by the synapsin condensates. The authors will need to use another method (e.g. using inert proteins with different hydrodynamic radii) to double check the approximate pore sizes of synapsin condensates.

We thank the Reviewer for raising this point. We have now performed additional reconstitution experiments by co-incubating synapsin-driven condensates with nanobodies and antibody conjugates. We used these probes as inert molecules, to test the penetration into the condensates. Neither the nanobodies, nor the antibodies recognizes any specific molecules within the condensates, as we used: (1) anti-mouse secondary nanobodies, and (2) a mixture of anti-LAMP1 rabbit antibodies and goat anti-rabbit secondary antibodies. When applied onto synapsin condensates alone, both nanobodies and the antibodies were able to penetrate within the condensates, and even enrich there. However, in synapsin/SV condensates, only nanobodies were enriched, while the antibody conjugates remained excluded and accumulated at the interface, probably due to non-specific interactions.

These data corroborate with our analysis using fluorescently-labeled dextran, and indicate that condensates act as molecular sieves, able to exclude molecules based on their size.

We include these experiments as **Supplementary Figure 4** in the revised version of the manuscript and amended the text (**lines 100-105**).

2. Ext. Data. Fig. 2, the workflow of sample loading does not match with the description in the method (supp. p5 lines 151-). When were SVs added? Before or after synapsin 1 condensation? The authors proposed that stronger FM 4-64 signal at the periphery was due to penetration of the dye (p4, lines

88-). A control experiment with FM4-64 dye on synapsin only condensates is needed. An alternative explanation by the authors is due to local uneven distribution of SVs, probably corresponding to the “outer area” as defined in Figure 2. This is an interesting point, as the boundaries in condensed droplets of phase separated systems are very interesting topic.

We have now amended the text in the Methods, to clarify that SVs were added once the condensates were formed. We have also included a control of synapsin condensates incubated with FM 4-64 dye (new panel b in Supplementary Fig. 2).

The question is how the MSD curve and mobility parameters of synapsin 1 in the outer area (or boundary area) would look like? Would these parameters be different from synapsins in the inner area? According to the experimental pipeline described in the manuscript, SV labelling with FM4-64 was simply achieved by mixing the dye with SVs followed by immediate imaging. How do authors control the labelling of SVs by the dye (e.g., how the experimental conditions were determined for optimal dye labelling)?

We performed the quantitative analysis of synapsin diffusion using their single-molecule trajectories projected onto the two-dimensional xy plane. To avoid the strong edge effect of the spherical condensates, we only analyzed movements of synapsin 1 molecules located >500 nm away from the edge (core regions of the condensates).

When analyzing the edge region, we have indeed found slower diffusion, as shown in the figure below. However, it is difficult to conduct a definitive quantitative assessment of the relative importance of the edge effect, due to the analysis procedure, versus the true slowing of diffusion near the condensate surface, which is why we also omit showing this data in the manuscript. In the revised manuscript, we further clarified this point in lines 140-142 in the main text and the subsection “*Analysis of the MSD- Δt plots obtained from ultrafast SMT of Halo7-synapsin 1 in the condensates in vitro*” in Methods).

Figure 1 | Quantitative analysis of synapsin diffusion taking into consideration the edge region.

3. Figure 2, the authors used the dwell time to describe the exchange properties but without any method description. What are the criteria for defining molecules “newly arrived” at the condensate and “left” the condensate? Also, despite the photobleaching lifetime of the fluorescent molecules, molecules diffuse out of focal plane will contribute to the signal loss during tracking the single molecule localizations. Can the authors add more details on this and discuss the influence of above-mentioned situations?

Thank you for the valuable comments about the results shown in the current, revised Fig. 2c (in the original manuscript, it was in Fig. 2b). As stated by Reviewer 1, before explaining our data about synapsin diffusion in the condensate, we wanted to first show that the exchange of synapsin molecules

in the condensate with those in the bulk solution is detectable by single-molecule imaging.

We have now strengthened the interpretations of the measurements and clarified the ambiguity of “newly arrived” and “departing” molecules. To address these problems, we have conducted new observations to measure the dwell lifetimes of synapsin 1 molecules in the condensates, using clear definitions of newly arrived and departing molecules, as described in the next paragraph and the following subsections (3-1) and (3-2) of the Reviewer’s comments. below. The revised manuscript now provides clearer explanations on the arrival and departure times of synapsin 1 molecules. Please see **lines 120-129 in the main text** and the subsection “**Determination of the dwell lifetime of Halo7-synapsin 1 (JF549 labeled) in the condensate in vitro**” in Methods.

In brief, we conducted our microscope observations using the oblique illumination of the home-built TIRF microscope by placing the focus to approximately 1~3 μm above the coverslip surface (the variation in the focal plane was the result of the adjustment made for each condensate) so that the equatorial planes of the condensates are within the focus (around ± 400 nm from the exact focal plane). In our observations, we paid particular attention to the two-dimensional movement/diffusion of synapsin 1 molecules on and near the xy focal plane (around ± 400 nm from the exact focal plane) assuming that their movements are isotropic in the core volume of the condensates (>500 nm away from the condensate surface).

(3-1) What are the criteria for defining molecules “newly arrived” at the condensate and “left” the condensate?

Newly arrived molecules include the following two types: (1) the molecules that arrived at the edges of the condensates from outside in/near the focal plane; we only selected the clear spots and thus probably those arriving approximately within ± 200 nm from the exact focal plane are likely considered (**see new Fig. 2b for typical trajectories**) and (2) those that came into clear focus (probably around ± 200 nm) within the condensate due to diffusion in z-direction. However, in the present study, we only analyzed the molecules categorized into class (1) as our intention was to examine diffusion projected onto the 2D xy plane, presupposing that synapsin 1 movements in the z-direction is the same as movement in x and y-directions. Furthermore, molecules of class (2) constitute less than ~5% of the total synapsin molecules.

(3-2) Also, despite the photobleaching lifetime of the fluorescent molecules, molecules diffuse out of focal plane will contribute to the signal loss during tracking the single molecule localizations.

The molecules that “departed” from the view field within the condensates include the following: (1) the molecules clearly observed to have departed from the condensate edges, that is, those whose departures from the condensate edges occurred in or near the focal plane (**see new Fig. 2b for typical trajectories**), (2) the molecules that disappeared due to photobleaching while remaining distinctly visible in the focal plane within the condensates, and (3) the molecules that diffused out of the focal plane in the z-direction ($> \pm 400$ nm from the center of the focal plane) within the condensates. The molecules that disappeared by the third process were excluded from our analysis, because we focused on the diffusion of synapsin 1 molecules projected onto the 2D xy plane, and for this, we assumed that synapsin 1 movements in the z-direction are congruent with those in x and y-directions. Furthermore, the number of synapsin 1 molecules in the condensates that disappeared due to the third process was small. Specifically, ~3% and ~7% of the total synapsin molecules we observed disappeared in this manner in the condensates with and without SVs, respectively, across an observation duration of 30 s (see the table below for precise counts).

The small numbers of arriving and departing molecules due to diffusion in the z-direction validate this analysis method, ensuring that the analysis results would not be skewed by the exclusion of rapidly diffusing molecules.

As described in **Methods**, the effect of photobleaching was corrected by the following equation.

$$\frac{1}{\tau_{\text{true}}} = \frac{1}{\tau_{\text{observed}}} - \frac{1}{\tau_{\text{bleach}}}$$

This correction for the directly observed events is essential for accounting for the distribution of the time windows for individual molecules due to the photobleaching process. Therefore, exclusions of the molecules that disappeared due to diffusion along the z-axis would not affect the photobleaching correction.

In the new measurements, we performed a time-lapse observation (15 Hz; every 66.7 ms, but the camera exposure time and laser illumination duration were 16.7 ms for each observation) to lower the effect of photobleaching. Under these observation conditions, the photobleaching lifetime was 54.4 s (described in **In 120-137 in the main text** and in the subsection of “*Determination of the dwell lifetime of Halo7-synapsin 1 (JF549 labeled) in the condensate in vitro*” in **Methods**). We observed 30 s instead of 20 s to ensure we did not overlook the molecules that remain longer in the condensates. The numbers of condensates we examined were increased (new **Fig. 2c**).

We found that the dwell time distributions could be fitted by the sum of two exponential functions. To ensure clarity, we now directly show the distributions of dwell durations for individual molecules rather than the cumulative histogram we employed previously. The shorter lifetimes observed remain comparable to the previous results (~0.7 s), while longer lifetimes were ~27 s, rather than the previous values of 6 to 8 s (new **Fig. 2c**). The new measurements were performed with clear definitions of “newly arrived” and “departing” molecules, following the Reviewer’s recommendations, and the observation durations employed were longer. Thus, the amended results should represent more accurate dwell lifetimes.

Table 1. Counts of synapsin 1 molecules arriving or leaving condensates in the SMT measurements.

Arrival			Departure		
	# of spots (%)			# of spots (%)	
	- SVs	+ SVs		- SVs	+ SVs
Case (1)	435 (94.6)	433 (95.0)	Case (1)	362 (83.2)	349 (80.6)
Case (2)	25 (5.4)	23 (5.0)	Case (2)	41 (9.4)	69 (15.9)
Total	460	456	Case (3)	32 (7.4)	15 (3.5)
			Total	435	433

4. Figure 2c, the authors showed two typical tracks of synapsin 1 with or without SVs. However, the cumulative displacement as well as the maximum MSD of the molecules are small, which indicate the molecules are confined or oscillate within a small region. Such motions do not seem to match properties of molecules that undergo more or less random walk in the condensed phase. Did they observe longer range displacements in their images? One would expect that at least for certain trajectories the range explored by synapsin 1 may cover several hundreds of nm or even at um level.

Can the authors show some example trajectories of molecules entering or leaving the condensed phase (as these trajectories will likely be much longer) (4-1)?

In the original manuscript, we presented typical trajectories spanning 50 frames for all three observation frequencies (60, 250, and 1,000 frames/s) to align with the x-axis of the MSD- Δt plots displayed in Fig. 2d (**now Fig. 2e in the revised manuscript**). However, we now understand that this presentation could be misleading. Therefore, in the revised manuscript, we show the trajectories of 300 frames (Fig. 2c in the original manuscript, which is now **Fig. 2d in the revised manuscript**). The color changes in the trajectories simply indicate the elapsed time (refer to the color bar on the right). Additionally, we have included longer trajectories as suggested by the Reviewer, **now presented in Fig. 2b**.

For quantitative assessment of the long-term behaviors of synapsin 1 diffusion, we calculated MSD- Δt plot for 300 steps (for 5 s) for the 60-Hz trajectories, now displayed in the **new Fig. 2f**. Furthermore, MSD- Δt plots for 50 frames were generated for the 15- and 30-Hz trajectories (for 3.3 and 1.67 s, respectively), by re-sampling the longer trajectories obtained at 60 Hz (**Fig. 2e**). The summary figure of the diffusion anomaly parameter α (**Fig. 2g**) is now plotted against “Observation durations,” which represents the durations of the full-scales of x-axes in the graphs shown in **Fig. 2e** (for 50 frames for different observation frequencies) because the anomaly parameters were obtained through fitting in this time scale. In addition, the updated figure (**Fig. 2g**) now includes the new data points obtained at 15 and 30 Hz (3.3 and 1.67 s, respectively). The main text **lines 143-184** and the subsection “**Analysis of the MSD- Δt plots obtained from ultrafast SMT of Halo7-synapsin 1 in the condensates in vitro**” in **Methods** have been revised accordingly.

The resolution (or precision/detection uncertainty) of the fluorescent signal is proportional to the square root of photon number and thus proportional to the square root of imaging frequency. How was the resolution (precision) distribution and the mean resolution of the tracks in different imaging frequency were defined (4-2)?

In the original manuscript, the employed laser intensities were summarized in the subsection “**In vitro ultrafast SMT of synapsin 1**” in **Methods**, and the localization precisions of the fluorescent probes attached to the glass (using glass-base dishes) we evaluated at three frame rates were shown in Extended Data Fig. 4a (**now Supplementary Fig. 5b**). In the revised manuscript, we made this point clearer by indicating that the localization precisions are shown in this figure panel, and, in addition, we also described the localization precisions (numerical values) in the same paragraph, while the employed laser intensities are described in the respective subsection in **Methods**.

If the resolution is about the same level of what the authors claimed, the motions of molecules may be masked by the detection uncertainty of the microscope. For example, at 1000 Hz, the authors observed even in the absence of SV, synapsin 1 only experienced a ~67 nm oscillation, which is almost the same level of the estimated signal resolution under the experiment condition. Thus, the molecule’s true diffusion could be much smaller than what the authors have observed, making the comparisons between different conditions not meaningful (4-3).

The diffusion coefficient and anomaly parameter were determined from the MSD- Δt plots shown in Fig. 2d in the original manuscript, which are **now presented in Fig. 2e**. These plots were the averages of the MSD- Δt plots calculated from 200–500 trajectories (as stated in the caption). We have now made it clear in the main text that these results were obtained from averaging the MSD- Δt plots of multiple trajectories **lines 143-145 in the main text**.

To further address the point raised by the Reviewer, we have also included the MSD- Δt plots for immobile fluorescent molecules (JF549) bound to glass in **Fig. 2e** (grey plots). Please also see a typical 60-Hz trajectory for an immobile JF549 molecule newly displayed in **Fig. 2d**. These would further clarify the differences of the plots (magenta, blue, and grey) in **Fig. 2e**. Overall, this analysis indicates that our technology is able to differentiate well between different mobility conditions.

5. Figure 2d & f, the authors used anomalous diffusion model to analyse the MSD. Oddly, the diffusion anomaly parameter α decreased with imaging frequency (p6 line 149). The diffusion anomaly parameter should be independent of the sampling frequency. The fact that the anomaly parameter changes with detection frequency further indicates that the observed diffusion may be masked by the detection uncertainty (5-1). Another odd result is the control MSD curve of “on glass” group are with extremely low MSDs and sometimes even with negative values (especially in 1000 Hz). Since mean square displacement should always be a non-negative value, can the author elaborate why the negative MSD values were presented? As mentioned before, the detection uncertainty of the signal will create artificial displacements, and the level is about $2 \times 20 \times 20 \text{ nm}^2 \sim 0.001 \text{ um}^2$ even with a resolution (precision) as high as 20 nm. If only consider the contribution of photon numbers, the resolution in 1000 Hz should be at least 4 times to that in 60 Hz, the contribution of MSD would be $\sim 0.01 \text{ um}^2$, which is much higher than what the authors showed in Figure 2d. Can the authors explain why the control MSD values are so low (5-2)?

We now realize that our x-axis in Fig. 2f in the original manuscript (**now Fig. 2g in the revised manuscript**) could have been confusing for readers. As we present the results for the 50-frame data in Fig. 2d (**now Fig. 2e**), the inverse of the frame rate is proportional to the total observation duration in time. To clarify this, we have explicitly labeled the x-axis as “Observation duration (s)” in **Fig. 2g in the revised manuscript**.

Encouraged by the comments of Reviewer 1, in the revised manuscript, we have expanded the explanation about the observed dependence of the anomaly parameter on the observation duration, as shown in **Fig. 2g (lines 171-187 in the main text)**. Such dependences on the observed time duration are related to the diffusion suppression as a function of the diffused area, namely, the reductions of α values for longer observation durations would indicate the presence of longer-range correlation lengths in the condensates than those covered here up to the observation time scale of 3.3 s. The argument is further advanced in the main text, suggesting that direct but multiple synapsin 1-synapsin 1 interactions propagate at least to a few tens of synapsin 1 molecules, which is consistent with the network mechanisms that protein condensates are considered to form, as described in Choi et al., 2020 [1].

Following a prevalent convention in the single-molecule diffusion field, the offset of the MSD- Δt plot was subtracted in the display shown in Fig. 2d in the original manuscript (**now Fig. 2e in the revised manuscript**). This offset is attributed to the single-molecule localization error, as suggested by the Reviewer. We agree with the Reviewer that we should show this explicitly in this manuscript. Therefore, for enhanced clarity, we have written the following equation in the subsection “**Analysis of the MSD- Δt plots obtained from ultrafast SMT of Halo7-synapsin 1 in the condensates in vitro**” in **Methods** (in the original manuscript, we omitted the constant C , which represents the offset), and explained that we subtracted the offset value from the MSD in the MSD- Δt plots). Namely:

$$\text{MSD} = \langle r^2(t) \rangle = 4D_\alpha t^\alpha + C ,$$

where r , t , D_α , α , and C are the displacement, elapsed time, diffusion coefficient, anomaly parameter, and offset (derived from the localization error). In addition, we have included raw MSD- Δt plots (without offset subtraction) in **Supplementary Fig. 5c**.

6. Figure 2 (in vitro reconstitution) concluded that the mobility of synapsin 1 was decreased in the presence of SV, and the authors speculated that “the interaction with synapsin 1 reciprocally slows the dynamics of synapsin by prolonging its dwell lifetimes in the condensates” (p6, lines 162-). But Ext. Data. Figs. 6 and 8 (cultured hippocampal neurons), the authors concluded that “synapsin confinement and diffusion rates are independent of SV” (supp. P12, line 309; p8, line 205). How should we understand the role of SV on synapsin 1 mobility?

In the reconstituted system we had conditions without or with SVs, and the SVs were found, as expected, to slow down synapsin mobility. However, this is very different that the analysis in neurons, where we image the motility of SVs by expressing synaptophysin-mEos3.2. Here, the expression levels of synaptophysin do not alter the number of available SVs, and we therefore concluded that synaptophysin expression does not change synapsin mobility. We rewrote the main text to clarify this aspect, **lines 265-274**.

7. Ext. Data. Fig. 8, the authors ectopically co-expressed synapsin 1 and intersectin in CV-1 cells and resulted in condensates formation and confinement of synapsin 1. The estimated D was $\sim 0.05 \mu\text{m}^2/\text{s}$, which is comparable to D of synapsin 1 in hippocampal neurons (Figure 3). Presumably CV-1 cells do not contain SV like structures, does this mean the confinement of synapsin 1 is an intrinsic property of the molecules within a condensate (i.e., independent of SVs as the authors have stated)? Additionally, both D values obtained in neurons and in CV-1 cells are much higher than those derived from the in vitro reconstituted system (Figure 2, in the absence of SV, $\sim 0.02 \mu\text{m}^2/\text{s}$ for 1000 Hz, and ~ 0.008 for 60 Hz), how to reconcile these results obtained from different systems?

We now included additional measurements by co-expressing synapsin 1 with its interaction partners Grb2 and synaptophysin (**new Supplementary Fig. 9**). The diffusion coefficient seems to be very similar in all these systems, which we report in the main text (**lines 257-261**), while remaining cautious to extrapolate this as a general feature of synapsins.

The differences in absolute values between reconstituted systems and the measurements in the complex cellular environment might result from the distinct chemical potential of synapsin driven by its varying phosphorylation status inside the cytoplasm or presence of interaction partners within the cell (i.e., the proline-rich motifs of synapsin could interact with many SH3 domain-containing proteins).

8. P6, lines 159: the authors mentioned “fraction of the volume”. Do authors have a way to estimate the volume fraction of synapsin 1 and SV? When the authors used “3 nM” to describe the SV concentration, how did they determine the concentration of SV? Each SV is associated with hundreds of proteins (PMID: 17110340)¹, then 3 nM SV may correspond to a few μM proteins. EM graphs clearly showed that majority space of a bouton (or the reserved pool SV area) is occupied by the SVs. Thus, the volume fraction of SVs should be considered together with synapsin 1.

We use the published fluorescence correlation spectroscopy data (i.e., Takamori et al., 2006) to relate the weight/volume of SV proteins to the copy number of SVs. This is now clarified in **Methods**. The molar ratio and average number of SVs in central synapses are determined according to Wilhelm et al., 2014.

9. Figure 3 and 4: the authors found that synaptophysin (as the mark for SVs) has a similar diffusion property compared to synapsin 1. Does this mean a comparable mobility of SV and synapsin 1? This is counterintuitive, as one would expect that SVs move much slower than synapsins. Furthermore, the authors treat synaptophysin movement same as SV movement. Synaptophysin may undergoes redistribution within an SV and move along with the SV macroscopically. How to differentiate these two types of motions of synaptophysin? It will be critical to carefully analyse trajectories of

synaptophysin. One would expect that both step sizes and overall MSD of synaptophysin should be <50 nm, which is about the size of an SV. Again, with such short step size, the imaging resolution becomes to be a critical factor.

We fully agree that the apparent diffusion of synaptophysin may result from its oligomerization at the surface of SVs, the pronounced membrane fluidity of SV bilayer, and the motion of SV as a whole. Here, we think the similarity in dynamics between synapsin and synaptophysin inside boutons is plausible as synapsin also interacts with the lipid bilayer of SVs. If there is a very fast-moving fraction of synapsin present in the condensate (that is not interacting with SVs directly) or a fast lateral diffusion of synaptophysin within a bilayer, we might not be able to capture these, due to technical limitations stemming from the observation speed we use. Given that the spatial resolution achieved in the live tracking experiments (~15 nm) is in the same range as the average radius of SVs (~20 nm²), it is not possible to distinguish between the movement of individual synaptophysin molecules in the SV membrane and the overall movement of SVs. While dissecting the precise individual contributions remains challenging in living neurons, at the temporal and spatial resolution of our experiments, synaptophysin motility remains a sufficient readout of the relative mobility of SVs. In the revised manuscript, we include this temporal and spatial limit of our experiments (**lines 223-229**).

10. Figure 4: it's surprising that syn1-IDR was sufficient to rescue SV distribution and SV release. Here the IDR in this manuscript covers synapsin 1 D and E domains, and the domain D contributes to the majority of the length and governs the phase separation property (ref 10)². It is surprising that the major SV binding domains (C and A) are not involved in synapsin 1's function. If the authors wish to claim that synapsin 1-D and E domain are sufficient to restore SV defects, additional experimental evidence, including EM images of recovered SV distributions and in vitro reconstitution of synapsin 1-DE recruitment of SVs are required. In addition, an early synapsin-TKO study suggested that synapsin 2, but NOT synapsin 1, can rescue synaptic depression in glutamatergic synapses (PMID: 18945891)³, how to reconcile these results with the current finding in the manuscript?

We agree about the novelty of that the finding that syn1-IDR was sufficient to rescue SV distribution and SV release was new and exciting. We have now performed the following experiments, to scrutinize this observation and ensure the robustness of this finding:

(i) we purified syn1-IDR and performed in parallel the reconstitutions of SVs with either synapsin 1 full-length (syn1-FL) or syn1-IDR. These data, presented in **the new Supplementary Fig. 11** in the revised manuscript, clearly indicate that syn1-IDR is sufficient to sequester SVs in vitro.

(ii) To test the proper targeting of both syn1-FL and syn1-IDR, we imaged synapsin triple-knockout (SynTKO) neurons transfected with these constructs. The data, shown **in the new Fig. 4a** of the revised manuscript, show that both constructs are targeted to boutons.

(iii) 3D-STED imaging using nanobodies against synaptotagmin-1, a bona fide SV protein, confirms that the accumulation of SVs is rescued in SynTKO neurons transfected with syn1-FL or syn1-IDR. This is now shown in **the new Supplementary Fig. 12** of the revised manuscript.

These experiments thus strengthen the notion that syn1-IDR is sufficient to rescue SV distribution and release. All these new data are included **in the Results (lines: 265-274 & 278-286)** of the revised manuscript. Importantly, we also emphasize (**lines 298-303**) that other regions of synapsin 1 and different synapsin isoforms may play a central role in regulating neurotransmitter release across different stimulation frequencies or in different brain regions/types of synapses.

11. Also in Figure 4: many studies including some from the senior author have demonstrated that synapsin TKO led to drastic SV loss, particularly the reserve pool. Considering different pools of SVs have different mobilities (PMID: 22442064, 20643088) ^{4, 5}, how the authors evaluate their measured SV mobilities in boutons of WT and synapsin-TKO synapses?

We analyzed single-molecule localization of synaptophysin-mEos3.2 in wild-type and synapsin triple knockout neurons, as well as in rescues with syn1-FL and syn1-IDR. The data, shown **in the new Supplementary Fig. 10**, indicate significantly greater dispersion of synaptophysin (SV) localizations up to 2 μm -away from synaptic boutons of SynTKO neurons. This is in line with the perturbation in diffusion behavior of releasable and reserve pools of SVs, as indicated in the previous studies, Kamin et al., 2010 [3]; Orenbuch et al., 2012 [4]. This this phenotype was rescued upon re-introducing either syn1-FL or syn1-IDR. In addition to including these new data in the revised manuscript, we elaborate the plausible mechanisms **in the Discussion section (lines 328-337)**.

Reviewer #2 (Remarks to the Author):

Hoffmann and co-workers present a microscopy-based study on highlighting how the protein synapsin controls sequestering and dynamics of extracellular vesicles through condensation (or liquid-phase separation). They show this for in vitro and cell measurements as well as link it to signalling events. Many controls are included, which makes this story very convincing. These are important insights with high impact. This certainly deserves publication.

We thank the Reviewer for the positive assessment of our work, the careful reading and the constructive suggestions, which we integrated in the revised manuscript.

One general issue though: The authors are very fixed on LLPS. However, modification of EVs by synapsin has also been indicated through other processes such as aggregation of synapsin. The transition (or distinguishing) between aggregation and LLPS is any way very smooth. Can the authors at all comment on this, maybe the data could be described through a more unifying instead of contradictory model?

We fully agree that the precise mechanism of how synapsins preserve the vesicle cluster is still under debate, which has actually motivated our study. The likely mechanisms are: (1) synapsins crosslink the vesicles, acting as tethers; (2) synapsins form a liquid phase, capturing synaptic vesicles in it; or (3) a mixture of both, since these mechanisms are not mutually exclusive.

We have now synthesized the main arguments into the following text: *“Initially, it was proposed that synapsins act as tethers by crosslinking the vesicles together (Hirokawa et al., 1989 [5]). This implies that synapsins play a role in physically connecting the vesicles, thereby maintaining their clustered arrangement. The high local concentration of synapsins (Wilhelm et al., 2014 [6]) suggests them being more than a mere crosslinkers of SVs (Milovanovic et al., 2017 [7]). We and the other showed that synapsins can form a liquid phase, effectively ensnaring the synaptic vesicles within it (Milovanovic et al. 2018, [8]; Pechstein et al., 2020 [9], Hoffmann et al., 2021 [10]). In this scenario, synapsins act as a cohesive medium, keeping the vesicles together in a clustered state while allowing for their high motility (Kamin et al., 2010 [3]; Orenbuch et al., 2012 [4]; Joensuu et al., 2016 [11]). It is also plausible that a combination of both mechanisms occurs, as they are not necessarily mutually exclusive (Zhang et al., 2021 [12]).”*

This is now integrated **in the Discussion section** of the revised manuscript (**lines 328-337**).

I have a few small comments:

(1) Page 4, line 82: How large are the condensates?

We have now quantified the condensate size and included this in the **Supplementary Fig. 1g**.

(2) General to the diffusion data (whether in vitro or in cells): Did the authors check for free dye, unbound to the Halo-Tag? Maybe show one example with another dye or highlight the diffusion for the free dye (as they did once)? Also, how did the authors include noise in the MSD data analysis?

For the in vitro measurements, we performed additional experiments and measured the diffusion of JF549-Halo ligand (non-binding) within the synapsin 1 condensates (**lines 207-212 in the main text and Supplementary Fig. 5d**). The diffusion coefficient and the diffusion anomaly parameter (α) were smaller in the condensates with SVs compared with those without SVs. However, these values were much larger than those for Halo7-synapsin. These results were consistent with the data shown in **Fig. 1** (dextran exclusion experiments). Taken together, these results demonstrate that synapsin condensates work as a molecular filter even for small molecules.

For the measurements in neurons, capturing the movement of unbound Janelia Fluor 635 (JF635) is unlikely with the frame rate used (100 Hz) in the in vivo experiments. In contrast to the fluorescent protein EGFP, the organic dye JF635 is much smaller and, therefore, much faster moving. An additional advantage is that JF635 is fluorogenic, so the unbound JF635 yields no fluorescence.

We have extensively explained this **in the revised Methods** (also described as an answer to point #5 of Reviewer 1).

(3) Page 5, lines 133ff: This sentence is not clear – why “even” and what “resulting confinement”?

It was meant that we could detect the effect of SVs “even” in short-term, short-range observations. To clarify, we totally revised the paragraph describing the results obtained in the time scale of 2–4 ms. **Please see lines 188-197 in the main text.**

(4) Page 5, line 140: How did the authors determine the radius of gyration or the diameter of the SVs?

The radius of gyration for synapsin 1 was taken from Alpha Fold and the diameter of SVs was adopted from Takamori et al., 2006 [2]. However, in the revised manuscript, this entire section has been rewritten for clarity, in **lines 198-206 of the main text.**

(5) Page 6, lines 152ff: I did not quite get how the authors got to this conclusion.

To clarify this point, we revised this part. The findings presented in **Fig. 2g** and **Fig. 2h**, considering that SVs occupy only a minor fraction of the volume in our reconstituted condensates, imply that SVs suppress the diffusion of synapsin 1 located on the SV surfaces through direct molecular interactions, and this slowing effect is propagated, exerting a global effect, likely due to the synapsin 1 interactions that are indeed responsible for the condensate formation. Since the SVs are sequestered into condensates by the interaction with synapsin 1 molecules, it would be reasonable to conclude that this interaction reciprocally slows the dynamics of synapsin 1 (**lines 188-197 in the main text**).

(6) Page 6, line 159: Give numbers for the molar ratios here.

This is now included (**lines 198-199 in the main text**).

(7) Page 6, lines 161ff: Where is the proof for this statement?

The longer dwell lifetime and slowed diffusion might be induced by the same underlying molecular interactions, but their relationships could be quite complicated. To omit an inadvertent simplification, we took out this statement.

(8) General: Why and how did the authors use PALM? This does decrease the temporal resolution for single-molecule tracking it seems. Why did they not just use a low enough labelling concentration?

For single molecule localizations in neurons, we used low labeling concentrations in the case of Janelia Fluor 635 (100 pM– 500 pM) and low photoactivation rate for mEos3.2 (i.e., photoswitching occurs at ns, below the sampling time of our experiments), which ultimately also leads to a low labeling concentration. The low photoactivation rate in PALM was more convenient than simply using low concentrations of proteins, since it was far easier to fine-tune and manipulate, enabling faster experimentation. To avoid confusion, in the revised manuscript, we have updated **the main text (lines 216-217)** and **figure legends (lines 510 and 512)** to use '*single molecule localizations*.'

(9) Page 7, line 175: Quantify the enrichment!

We quantified the enrichment and updated the main text and figure legends accordingly (**lines 234-235 and 518 in the revised manuscript**). There is a ~3-fold enrichment inside synaptic boutons (337 of tracks are inside boutons vs 116 outside boutons).

(10) Page 7, lines 177: I got a little bit confused here. Which population is slowed-down, which one is confined to certain regions and which one diffuses between boutons? My confusion mainly stems from the fact that confinement is used both for slow-down and confinement into regions. Also, always state numbers for alpha.

We now rewrote the text to explain the two populations (inside the bouton and outside the bouton) better and included both the diffusion coefficients and the coefficients of confinements for each of these two populations. Please see **lines 229-234 in the revised manuscript**.

(11) Page 9, lines 245ff: Maybe give comparing numbers for diffusion coefficients here?

In this particular case, we intentionally report the numbers separately. This is because the total number of tracks is purposely not adding up. In the first analysis approach, which is according to the localization (i.e., the regions within boutons vs. the regions outside of boutons), we consider all the tracks (Fig. 3b-e). In the second analysis approach (Supplementary Fig. 8), the filtering of the tracks is according to coefficient of confinement ($0 \leq \alpha \leq 0.75$) vs super-diffuse ($1.25 \leq \alpha \leq 2$). Thus, the tracks of intermediate confinement including the Brownian motion ($0.75 \leq \alpha \leq 1.25$) are filtered out completely. Importantly, despite this completely different analysis approach, the confined tracks are localized inside boutons, while the super-diffusive ones are between boutons, as we would expect, according to our general hypothesis. This is now explained **in Methods (In 268-275)**.

(12) Page 10, line 280: What is meant by fast here? Difficult to get a feeling for it after all the discussed confinement.

We apologize for the confusion and have rewritten text to clarify that our data indicate synapsin molecules to remain highly diffusive both *in vitro* ($0.019 \mu\text{m}^2/\text{s}$) and *in vivo*/in neurons ($0.051 \mu\text{m}^2/\text{s}$) despite their spatial confinement. The paragraph now reads: “Finally, both *in vitro* ($0.019 \mu\text{m}^2/\text{s}$) and *in living neurons* ($0.051 \mu\text{m}^2/\text{s}$), synapsins maintain their fast motility within these condensates despite their spatial confinement. This allows synapsin molecules to be rapidly phosphorylated by kinases upon the neuronal activity at the interface of condensates without the need for enzymes to fully penetrate into condensates. Thus, the size of meshwork coupled to the internal dynamics of proteins within a condensate provides an additional layer of specificity and allows for SV condensates to act as buffers of proteins and enzymes.” Please see **lines 343-348 in the revised manuscript**.

References

1. Choi, J.-M., Holehouse, A. S. & Pappu, R. V. Physical Principles Underlying the Complex Biology of Intracellular Phase Transitions. *Annu Rev Biophys* 49, 1–27 (2020).
2. Takamori, S. *et al.* Molecular anatomy of a trafficking organelle. *Cell* 127, 831–846 (2006).
3. Kamin, D. *et al.* High- and Low-Mobility Stages in the Synaptic Vesicle Cycle. *Biophys. J.* 99, 675–684 (2010).
4. Orenbuch, A. *et al.* Synapsin Selectively Controls the Mobility of Resting Pool Vesicles at Hippocampal Terminals. *J Neurosci* 32, 3969–3980 (2012).
5. Hirokawa, N., Sobue, K., Kanda, K., Harada, A. & Yorifuji, H. The cytoskeletal architecture of the presynaptic terminal and molecular structure of synapsin 1. *J. Cell Biol.* 108, 111–126 (1989).
6. Wilhelm, B. G. *et al.* Composition of isolated synaptic boutons reveals the amounts of vesicle trafficking proteins. *Science* 344, 1023–1028 (2014).
7. Milovanovic, D. & Camilli, P. D. Synaptic Vesicle Clusters at Synapses: A Distinct Liquid Phase? *Neuron* 93, 995–1002 (2017).
8. Milovanovic, D., Wu, Y., Bian, X. & Camilli, P. D. A liquid phase of synapsin and lipid vesicles. *Science* 361, 604–607 (2018).
9. Pechstein, A. *et al.* Vesicle Clustering in a Living Synapse Depends on a Synapsin Region that Mediates Phase Separation. *Cell Reports* 30, 2594–2602.e3 (2020).
10. Hoffmann, C. *et al.* Synapsin Condensates Recruit alpha-Synuclein. *J Mol Biol* 166961 (2021) doi:10.1016/j.jmb.2021.166961.
11. Joensuu, M. *et al.* Subdiffractional tracking of internalized molecules reveals heterogeneous motion states of synaptic vesicles. *J Cell Biol* 215, 277–292 (2016).
12. Zhang, M. & Augustine, G. J. Synapsins and the Synaptic Vesicle Reserve Pool: Floats or Anchors? *Cells* 10, 658 (2021).

REVIEWERS' COMMENTS

Reviewer #1 (Remarks to the Author):

The authors have added more experiments and revised texts to address most of this reviewer's questions and concerns. The manuscript has been significantly improved. I only have a few minor concerns for the authors to consider. I am not asking for more experiments. Instead, the authors should consider to elucidate or discuss these points in appropriate places in the manuscript.

1. What is the biophysical meaning for the decreased α values along different observation time points?

2. The unit of diffusion coefficients in all anomalous diffusion were dimensionally incorrect, since the $MSD=4Dt^\alpha$, the diffusion coefficient (D) should have a unit of $\mu m^2/s^\alpha$ instead of $\mu m^2/s$. If the units among diffusion coefficients are different, directly comparing these values would not be appropriate. The authors should change these units in the manuscript as well as in the figures.

3. There is no apparent synapsin MSD difference in neurons (with SVs) and CV-1 cells (without SVs). This point should be discussed.

4. There is no apparent MSD difference between synapsin and synaptophysin (representing SVs). This point should also be discussed.

Reviewer #2 (Remarks to the Author):

The authors have well commented to all my previous concerns and excellently revised the manuscript accordingly. I have no further concerns and strongly suggest publication in Nature Communications.

**POINT-BY-POINT RESPONSE TO:
SYNAPSIN CONDENSATION CONTROLS SYNAPTIC VESICLE SEQUESTERING AND DYNAMICS**

REVIEWERS' COMMENTS

REVIEWER #1 (REMARKS TO THE AUTHOR):

The authors have added more experiments and revised texts to address most of this reviewer's questions and concerns. The manuscript has been significantly improved. I only have a few minor concerns for the authors to consider. I am not asking for more experiments. Instead, the authors should consider to elucidate or discuss these points in appropriate places in the manuscript.

We thank the Reviewer for the positive assessment of our work, the careful reading and the constructive suggestions for further discussion, which we all integrated into the revised manuscript.

1. What is the biophysical meaning for the decreased α values along different observation time points?

This is now clarified in the main manuscript file, In 181-185; In 189-193; In 206-209.

2. The unit of diffusion coefficients in all anomalous diffusion were dimensionally incorrect, since the $MSD=4Dt^\alpha$, the diffusion coefficient (D) should have a unit of $\mu m^2/s^\alpha$ instead of $\mu m^2/s$. If the units among diffusion coefficients are different, directly comparing these values would not be appropriate. The authors should change these units in the manuscript as well as in the figures.

This is now corrected for Supplementary figure 5a, d. The diffusion coefficients that appeared in all other figures are those evaluated and calculated from the first four MSD steps (i.e., the linear range, which implies $\alpha = 1$), so their dimensions are $\mu m^2/s$.

3. There is no apparent synapsin MSD difference in neurons (with SVs) and CV-1 cells (without SVs). This point should be discussed.

This is now discussed in the main manuscript file, In 266-273.

4. There is no apparent MSD difference between synapsin and synaptophysin (representing SVs). This point should also be discussed.

This is now discussed depth in the main manuscript file, In 232-239.

REVIEWER #2 (REMARKS TO THE AUTHOR):

The authors have well commented to all my previous concerns and excellently revised the manuscript accordingly. I have no further concerns and strongly suggest publication in Nature Communications.

We thank the Reviewer for the shared enthusiasm about our work and, especially, for the constructive and fruitful revision process.